# THE IMPLICIT BIAS OF MINIMA STABILITY IN MULTIVARIATE SHALLOW ReLU NETWORKS

**Mor Shpigel Nacson**[*] **& Rotem Mulayoff**[*]
Electrical & Computer Engineering, Technion

**Greg Ongie**
Mathematical and Statistical Sciences
Marquette University

**Tomer Michaeli & Daniel Soudry**
Electrical & Computer Engineering, Technion

## ABSTRACT

We study the type of solutions to which stochastic gradient descent converges when used to train a single hidden-layer multivariate ReLU network with the quadratic loss. Our results are based on a dynamical stability analysis. In the univariate case, it was shown that linearly stable minima correspond to network functions (predictors), whose second derivative has a bounded weighted $L^1$ norm. Notably, the bound gets smaller as the step size increases, implying that training with a large step size leads to 'smoother' predictors. Here we generalize this result to the multivariate case, showing that a similar result applies to the Laplacian of the predictor. We demonstrate the tightness of our bound on the MNIST dataset, and show that it accurately captures the behavior of the solutions as a function of the step size. Additionally, we prove a depth separation result on the approximation power of ReLU networks corresponding to stable minima of the loss. Specifically, although shallow ReLU networks are universal approximators, we prove that *stable* shallow networks are not. Namely, there is a function that cannot be well-approximated by stable single hidden-layer ReLU networks trained with a non-vanishing step size. This is while the same function can be realized as a stable two hidden-layer ReLU network. Finally, we prove that if a function is sufficiently smooth (in a Sobolev sense) then it can be approximated arbitrarily well using shallow ReLU networks that correspond to stable solutions of gradient descent.

## 1 INTRODUCTION

Neural networks (NNs) have been demonstrating phenomenal performance in a wide array of fields, from computer vision and speech processing to medical sciences. Modern networks are typically taken to be highly overparameterized. In such setting, the training loss usually has multiple global minima, which correspond to models that perfectly fit the training data. Some of those models are clearly sub-optimal in terms of generalization. Yet, the training process seems to consistently avoid those bad global minima, and somehow steer the model towards global minima that generalize well. A long line of works attributed this behavior to "implicit biases" of the training algorithms, *e.g.*, (Zhang et al., 2017; Gunasekar et al., 2017; Soudry et al., 2018; Arora et al., 2019).

Recently, it has been recognized that a dominant factor affecting the implicit bias of gradient descent (GD) and stochastic gradient descent (SGD), is associated with dynamical stability. Roughly speaking, the dynamical stability of a minimum point refers to the ability of the optimizer to stably converge to that point. Particular research efforts have been devoted to understanding *linear stability*, namely the dynamical stability of the optimizer's linearized dynamics around the minimum (Wu et al., 2018; Nar & Sastry, 2018; Mulayoff et al., 2021; Ma & Ying, 2021). For GD and SGD, it is well known that a minimum is linearly stable if the loss terrain is sufficiently flat w.r.t. the step size $\eta$.

Concretely, a necessary condition for a minimum to be linearly stable for GD and SGD is that the top eigenvalue of the Hessian at that minimum point be smaller than $2/\eta$ (see Sec. 2). Although this

---

[*]Indicates equal contribution. Correspondence: `{mor.shpigel,rotem.mulayof}@gmail.com`.

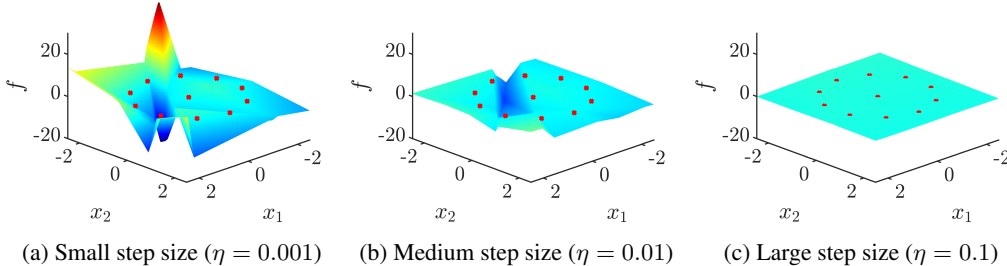

(a) Small step size ($\eta = 0.001$)    (b) Medium step size ($\eta = 0.01$)    (c) Large step size ($\eta = 0.1$)

Figure 1: **Larger step size leads to smoother prediction function.** We train a single hidden-layer ReLU network on a regression task with two-dimensional data, depicted by red points. The different panels show the predictor function $f$ obtained when training with different step sizes.

condition only characterizes the linearized dynamics, it has been empirically shown to hold in real-world neural-network training (Cohen et al., 2020; Gilmer et al., 2022). The linear stability condition turns out to have a strong effect on the nature of the network that is obtained upon convergence, both in terms of the end-to-end predictor function (Mulayoff et al., 2021), and in terms of the way this function is implemented by the network (Mulayoff & Michaeli, 2020).

Mulayoff et al. (2021) studied how linear stability affects a single hidden-layer univariate ReLU network, when trained with the quadratic loss. They showed that in this setting, stable solutions of SGD with step size $\eta$ correspond to functions $f$ satisfying

$$\int_{\mathbb{R}} |f''(x)|\, g(x)\mathrm{d}x \leq \frac{1}{\eta} - \frac{1}{2}, \tag{1}$$

where $f$ denotes the network input-output function, and $g$ is a weight function that depends only on the training data. This result implies that for univariate shallow ReLU networks, SGD is biased towards 'smooth' solutions[1]. Moreover, the larger the step size $\eta$, the smoother the solution becomes.

In this paper, we study the stable solutions of single hidden-layer ReLU networks with multidimensional inputs, trained using SGD and the quadratic loss. Particularly, in Sec. 3 we generalize the result of Mulayoff et al. (2021) to the multivariate setting. As it turns out, the natural extension of (1) involves the Radon transform of the Laplacian of the predictor function, $\Delta f$ (see Thm. 1). However, we show this result can also be interpreted in primal space as

$$\int_{\mathbb{R}^d} |\Delta f(\boldsymbol{x})|\rho(\boldsymbol{x})\mathrm{d}\boldsymbol{x} \leq \frac{1}{\eta} - \frac{1}{2}, \tag{2}$$

where $\rho$ is some weighting function. Thus, stable solutions of SGD in the multivariate case also correspond to smooth predictors (*i.e.*, functions whose Laplacian has a small weighted $L^1$ norm). The larger the step size, the smoother the function becomes. Figure 1 illustrates this phenomenon.

Additionally, we study the approximation power of single hidden-layer ReLU networks corresponding to stable minima. It is well known that shallow ReLU networks can approximate any continuous function over a compact set (Pinkus, 1999). However this does not imply that SGD can stably converge to such approximations. If there exist functions whose approximations are all unstable, then this property may be of limited practical interest. In Sec. 4 we prove that every convergent sequence of stable networks has a limit function that also satisfies the stability condition (Thm. 1). Building on this, we prove a depth separation result. Specifically, we show that there exists a function that does not satisfy the stability condition for any positive step size. Namely, it cannot be stably approximated by a single hidden-layer ReLU network trained with a non-vanishing step size. Yet, the same function can be realized as a two hidden-layer ReLU network corresponding to a stable minimum. Moreover, in Sec. 5 we show that if a function is sufficiently smooth (Sobolev) then it can be approximated arbitrarily well using single hidden-layer ReLU networks that correspond to stable solutions of GD.

Finally, in Sec. 3.3 and 6 we demonstrate our results. Particularly, we illustrate how our stable minima characterization (Thm. 1) can be used to predict certain properties of the solution. For example, for

---

[1]In a slight abuse of terms, in this paper we say a function is 'smooth' if some weighted $L^1$ norm of its second derivative is bounded.

certain isotropic data (*e.g.*, Gaussian), we show that a large step size tends to increase the biases of all neurons. We also demonstrate on the MNIST dataset the tightness of our stability bound, and that it predicts well the dependence of the stability and generalization performance on the step size.

## 2 BACKGROUND: MINIMA STABILITY OF SGD

In this section we give a brief survey on minima stability. Let us consider the problem of minimizing an empirical loss using SGD. We are interested in the typical regime of overparameterized models. In this setting, there exist multiple global minimizers of the loss. Yet, SGD cannot stably converge to any minimum. The stability of a minimum is associated with the dynamics of SGD in its vicinity. Specifically, a minimum is said to be stable if once SGD arrives near the minimum, it stays in its vicinity. If SGD repels from the minimum, then we say that it is unstable.

Formally, let $\ell_j : \mathbb{R}^d \to \mathbb{R}$ be differentiable almost everywhere for all $j \in [n]$. Here we consider a loss function $\mathcal{L}$ and its stochastic analogue,

$$\mathcal{L}(\boldsymbol{\theta}) = \frac{1}{n} \sum_{j=1}^{n} \ell_j(\boldsymbol{\theta}) \quad \text{and} \quad \hat{\mathcal{L}}_t(\boldsymbol{\theta}) = \frac{1}{B} \sum_{j \in \mathfrak{B}_t} \ell_j(\boldsymbol{\theta}), \tag{3}$$

where $\mathfrak{B}_t$ is a batch of size $B$ sampled at iteration $t$. We assume that the batches $\{\mathfrak{B}_t\}$ are drawn uniformly from the dataset, independently across iterations. SGD's update rule is given by

$$\boldsymbol{\theta}_{t+1} = \boldsymbol{\theta}_t - \eta \nabla \hat{\mathcal{L}}_t(\boldsymbol{\theta}_t), \tag{4}$$

where $\eta$ is the step size. Analyzing the full dynamics of this system is intractable in most cases. Therefore, several works studied the behavior of this system near minima using linearized dynamics (Wu et al., 2018; Ma & Ying, 2021; Nar & Sastry, 2018; Mulayoff et al., 2021), which is a common practice for characterizing the stability of nonlinear systems.

**Definition 1** (Linear stability). *Let $\boldsymbol{\theta}^*$ be a twice differentiable minimum of $\mathcal{L}$. Consider the linearized stochastic dynamical system*

$$\boldsymbol{\theta}_{t+1} = \boldsymbol{\theta}_t - \eta \left( \nabla \hat{\mathcal{L}}_t(\boldsymbol{\theta}^*) + \nabla^2 \hat{\mathcal{L}}_t(\boldsymbol{\theta}^*)(\boldsymbol{\theta}_t - \boldsymbol{\theta}^*) \right). \tag{5}$$

*Then $\boldsymbol{\theta}^*$ is $\varepsilon$ linearly stable if for any $\boldsymbol{\theta}_0$ in the $\varepsilon$-ball $\mathcal{B}_\varepsilon(\boldsymbol{\theta}^*)$, we have $\limsup\limits_{t \to \infty} \mathbb{E}[\|\boldsymbol{\theta}_t - \boldsymbol{\theta}^*\|] \leq \varepsilon$.*

Namely, a minimum is $\varepsilon$ linearly stable if once $\boldsymbol{\theta}_t$ enters an $\varepsilon$-ball around the minimum, it ends up at a distance no greater than $\varepsilon$ from it in expectation. Under mild conditions, any stable minimum of the nonlinear system is also linearly stable (Vidyasagar, 2002, p. 268). We have the following condition.

**Lemma 1** (Necessary condition for linear stability (Mulayoff et al., 2021, Lemma 1)). *Consider SGD with step size $\eta$, where batches are drawn uniformly from the training set, independently across iterations. If $\boldsymbol{\theta}^*$ is an $\varepsilon$ linearly stable minimum of $\mathcal{L}$, then*

$$\lambda_{\max} \left( \nabla^2 \mathcal{L}(\boldsymbol{\theta}^*) \right) \leq \frac{2}{\eta}. \tag{6}$$

This condition states that stable minima of SGD are flat w.r.t. the step size. Although this result was proved for the linearized dynamics, it was observed to hold also in practice, where the full nonlinear dynamics apply. Particularly, much empirical evidence on real-world neural-network training (Cohen et al., 2020; Gilmer et al., 2022) points out that GD and SGD converge only to linearly stable minima, *i.e.*, minima satisfying (6). More on dynamical stability and its interaction with common practices (*e.g.*, learning rate decay, absence of $\varepsilon$ and $B$ in Lemma 1 result, *etc.*) in App. A.

## 3 LARGE STEP SIZE BIASES TO SMOOTH FUNCTIONS

Consider the set of multivariate functions over $\mathbb{R}^d$ that can be implemented by a single hidden-layer ReLU network with $k$ neurons,

$$\mathcal{F}_k \triangleq \left\{ f : \mathbb{R}^d \to \mathbb{R} \ \middle| \ f(\boldsymbol{x}) = \sum_{i=1}^{k} w_i^{(2)} \sigma \left( \boldsymbol{x}^\top \boldsymbol{w}_i^{(1)} + b_i^{(1)} \right) + b^{(2)} \right\}, \tag{7}$$

where $\sigma(\cdot)$ denotes the ReLU activation function. Each $f \in \mathcal{F}_k$ is a piecewise linear function with at most $k$ knots[2]. Given some training set $\{\boldsymbol{x}_j, y_j\}_{j=1}^{n}$, we are interested in functions that globally

---

[2]A 'knot' is a boundary between two pieces (*i.e.*, intersection between hyperplanes). See Fig. 2 for illustration.

minimize the quadratic loss[3]

$$\mathcal{L}(f) \triangleq \frac{1}{2n} \sum_{j=1}^{n} \left( f(\boldsymbol{x}_j) - y_j \right)^2. \tag{8}$$

**Definition 2** (Solution). *A function $f \in \mathcal{F}_k$ is a 'solution' if $\mathcal{L}(f) = 0$, i.e., $f(\boldsymbol{x}_j) = y_j \; \forall j \in [n]$.*

We focus on the overparameterized regime ($kd > n$) in which there exist multiple solutions. We want to study the properties of solutions which correspond to stable minima of SGD. However, a key challenge is that any solution $f \in \mathcal{F}_k$ typically has infinitely many different parameterizations. In other words, there are various parameter vectors

$$\boldsymbol{\theta} \triangleq \begin{bmatrix} \boldsymbol{w}_1^{(1)\top} & \cdots & \boldsymbol{w}_k^{(1)\top} & \boldsymbol{b}^{(1)\top} & \boldsymbol{w}^{(2)\top} & b_2 \end{bmatrix}^\top \in \mathbb{R}^{(d+2)k+1}, \tag{9}$$

that can implement the same function $f$. Different parameterizations correspond to different minima, which may have different Hessian eigenvalues. Therefore, for a given step size $\eta$, some parameterizations of $f$ may be stable while others may not. Thus, to determine whether SGD can stably converge to a solution $f$, we need to check whether there exists *some* stable minimum $\boldsymbol{\theta}$, which corresponds to a parametrization of $f$. We therefore use the following definition.

**Definition 3** (Stable solution). *A solution $f \in \mathcal{F}_k$ is said to be stable for step size $\eta$ if there exists a minimum $\boldsymbol{\theta}^*$ of the loss that corresponds to $f$, where $\boldsymbol{\theta}^*$ is linearly stable for SGD with step size $\eta$.*

The next theorem characterizes stable solutions using the Radon transform $\mathcal{R}$ (see App. C) and the Laplace operator $\Delta$. Particularly, we use the inverse of the dual Radon transform, $(\mathcal{R}^*)^{-1}$, and interpret $\Delta f$ in the weak sense, *i.e.*, as a sum of weighted Dirac delta functions (see App. D).

**Theorem 1** (Properties of stable solutions). *Let $f$ be a linearly stable solution for SGD with step size $\eta$. Assume that the knots of $f$ do not coincide with any training point. Then*

$$\|f\|_{\mathcal{R},g} \le \frac{1}{\eta} - \frac{1}{2}, \tag{10}$$

*where $\|\cdot\|_{\mathcal{R},g}$ is the stability norm, defined as*

$$\|f\|_{\mathcal{R},g} \triangleq \int_{\mathbb{S}^{d-1} \times \mathbb{R}} \left| \left[ (\mathcal{R}^*)^{-1} \Delta f \right] (\boldsymbol{v}, b) \right| g(\boldsymbol{v}, b) \mathrm{d}s(\boldsymbol{v}) \mathrm{d}b, \tag{11}$$

*and $g(\boldsymbol{v}, b) \triangleq \min \left( \tilde{g}(\boldsymbol{v}, b), \tilde{g}(-\boldsymbol{v}, -b) \right)$ is a non-negative weighting function, with $\tilde{g}$ given by*

$$\tilde{g}(\boldsymbol{v}, b) \triangleq \mathbb{P}^2(\boldsymbol{X}^\top \boldsymbol{v} > b) \mathbb{E} \left[ \boldsymbol{X}^\top \boldsymbol{v} - b \middle| \boldsymbol{X}^\top \boldsymbol{v} > b \right] \sqrt{\left\| \mathbb{E} \left[ \boldsymbol{X} \middle| \boldsymbol{X}^\top \boldsymbol{v} > b \right] \right\|^2 + 1}. \tag{12}$$

*Here $\boldsymbol{X}$ is a random vector drawn from the dataset's distribution (i.e., sampled uniformly from $\{\boldsymbol{x}_j\}$).*

This theorem, whose proof is provided in App. E, shows that the step size constrains the stability norm of the solution. Notably, the constraint becomes stricter as the step size increases. Before interpreting this result, let us note that although it depends only on the step size, other hyper-parameters (*e.g.*, batch size, initialization) may potentially improve the bound. Yet, as we discuss in App. A, the effect of other hyper-parameters seems secondary in practical settings. The implications of Thm. 1 can be understood in primal space and in Radon space. In the following, we discuss both interpretations and give examples.

## 3.1 PRIMAL SPACE INTERPRETATION

Theorem 1 is stated in Radon space, which may be difficult to interpret. However, in some cases it can also be interpreted in primal space, by deriving an alternative form for the stability norm $\| \cdot \|_{\mathcal{R},g}$. Specifically, in App. G we show that if $g$ is piecewise continuous and $L^1$-integrable[4], then for all $f \in \mathcal{F}_k$ and $\rho = \mathcal{R}^{-1} g$ we have[5]

$$\|f\|_{\mathcal{R},g} = \int_{\mathbb{R}^d} |\Delta f(\boldsymbol{x})| \rho(\boldsymbol{x}) \mathrm{d}\boldsymbol{x}. \tag{13}$$

---

[3]We focus on MSE loss for simplicity, but the results can be extended to other loss functions, see App. J.

[4]Which is true, for example, when the training set is finite.

[5]This integral should be interpreted in the distributional sense (see App. G for details).

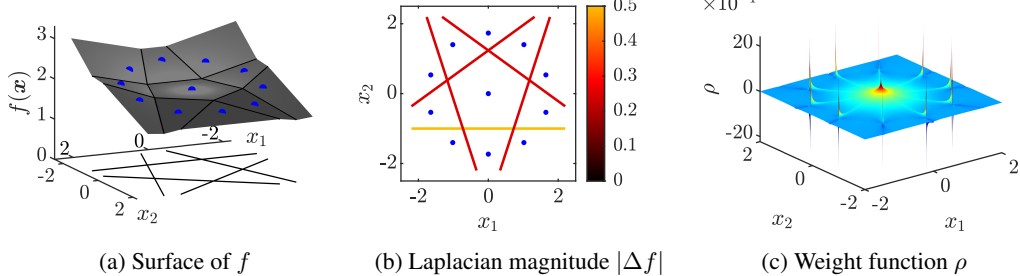

(a) Surface of $f$      (b) Laplacian magnitude $|\Delta f|$      (c) Weight function $\rho$

Figure 2: **Illustration of the stability norm.** Panel (a) depicts an interpolating function $f$. Panel (b) displays the absolute value of the Laplacian of $f$, *i.e.*, $|\Delta f|$. Here the color codes the amplitude of the delta functions. Panel (c) presents the weight function $\rho$. The stability norm is the weighted sum of line integrals of $\rho$, according to $|\Delta f|$.

In this presentation of the stability norm, $\rho$ is not necessarily non-negative. Nevertheless, all its hyper-plane integrals are non-negative, since $\mathcal{R}\rho = g \geq 0$. Thus, the stability norm can be interpreted as a non-negative linear combination of hyper-integrals of $\rho$ along the knots of $f$. This is visualized in Fig. 2. Hence, Thm. 1 combined with (13) implies that the larger the step size $\eta$, the smoother the solution becomes.

### 3.2 RADON SPACE INTERPRETATION

Another interesting interpretation of Thm. 1 can be derived in Radon space. First, let us examine how the weight function $g(\boldsymbol{v}, b)$ behaves as a function of $b$. For every fixed $\boldsymbol{v}$, the function $g(\boldsymbol{v}, \cdot)$ has a finite support, $[\min_j\{\boldsymbol{x}_j^\top \boldsymbol{v}\}, \max_j\{\boldsymbol{x}_j^\top \boldsymbol{v}\}]$. Moreover, $g(\boldsymbol{v}, \cdot)$ typically has most of its mass concentrated around the center of the distribution of the projected data points $\{\boldsymbol{x}_j^\top \boldsymbol{v}\}$, and it decays towards the endpoints (see *e.g.*, Fig. 3).

Next, let us interpret how the term $(\mathcal{R}^*)^{-1}\Delta f$ behaves. For a single hidden-layer ReLU network, $(\mathcal{R}^*)^{-1}\Delta f$ is a sum of Dirac deltas. Specifically, as shown in (Ongie et al., 2020), if $f$ is a function of the form $f(\boldsymbol{x}) = \sum_{i=1}^k a_i\sigma(\boldsymbol{v}_i^\top \boldsymbol{x} - b_i) + c$ with $\|\boldsymbol{v}_i\|_2 = 1$ for all $i \in [k]$, then (see App. F.3)

$$(\mathcal{R}^*)^{-1}\Delta f = \sum_{i=1}^k a_i\delta_{(\boldsymbol{v}_i, b_i)}, \tag{14}$$

where $\Delta f$ is the (distributional) Laplacian of $f$, and $\delta_{(\boldsymbol{v}, b)}$ denotes a Dirac delta centered at $(\boldsymbol{v}, b) \in \mathbb{S}^{d-1} \times \mathbb{R}$. We can thus define a parameter space representation for the stability norm as (see App. F.3)

$$S_{\boldsymbol{\theta}} \triangleq \sum_{i=1}^k |a_i|\, g\,(\boldsymbol{v}_i, b_i)\,. \tag{15}$$

Generally, this parametric representation of the stability norm satisfies $\|f\|_{\mathcal{R},g} \leq S_{\boldsymbol{\theta}}$, where equality happens whenever the ReLU knots of the representation do not coincide (*i.e.*, there is one Dirac function for each ReLU unit). Yet, this parametric view of the stability norm also obeys (see App. F.3)

$$S_{\boldsymbol{\theta}} \leq \frac{1}{\eta} - \frac{1}{2}\,. \tag{16}$$

Hence, larger step sizes $\eta$ push $S_{\boldsymbol{\theta}}$ to be smaller, and from (15) we see that $|a_i|$ will tend to be small. Also, since $g(\boldsymbol{v}, \cdot)$ typically decays towards the boundary of its support, this pushes the neurons' biases, $b_i$, away from the center of the distribution. The resulting effect is that the predictor function $f$ becomes flatter, especially near the center of the distribution. This is illustrated in Figs. 1 and 4(b).

### 3.3 EXAMPLES

Earlier we introduced two interpretations for the stability norm $\|\cdot\|_{\mathcal{R},g}$: one in primal space, which uses the weight function $\rho$, and one in Radon space, which uses the weight function $g$. In this section, we compute $g$ and $\rho$ for two toy examples, for which (13) holds.

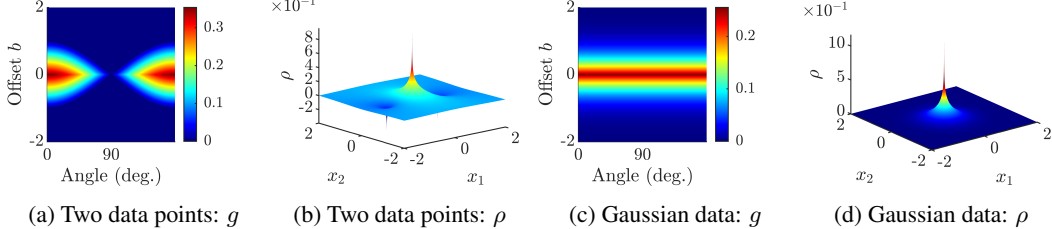

(a) Two data points: $g$    (b) Two data points: $\rho$    (c) Gaussian data: $g$    (d) Gaussian data: $\rho$

Figure 3: **Visualization of $g$ of Thm. 1 and $\rho$ of** (13) **for two toy examples.** (a), (b) Two data points $\boldsymbol{x}_1 = (1, 0)$ and $\boldsymbol{x}_2 = (-1, 0)$. (c),(d) Two dimensional Gaussian data, *i.e.*, $\boldsymbol{X} \sim \mathcal{N}(\boldsymbol{0}, \boldsymbol{I})$.

**Example 1: Two data points in $\mathbb{R}^2$.** Assume the dataset contains two points: $\boldsymbol{x}_1 = (1, 0)$ and $\boldsymbol{x}_2 = (-1, 0)$. In this case we can analytically calculate $g$ and $\rho$ (see App. H.1). Figure 3 depicts these functions. Here, $\rho$ has singularities at $\boldsymbol{x}_1$, $\boldsymbol{x}_2$ and at the origin. Yet, despite these singularities, all line integrals of $\rho$ are finite and thus the expression $\int_{\mathbb{R}^d} |\Delta f|(\boldsymbol{x}) \, \rho(\boldsymbol{x}) \mathrm{d}\boldsymbol{x}$ is well-defined. Moreover, while $\rho$ takes negative values, all its line integrals take positive values.

**Example 2: Isotropic distribution.** Suppose the data is isotropically distributed, *i.e.*, $\mathbb{P}(\boldsymbol{X}^\top \boldsymbol{v} > b)$ does not depend on the direction of $\boldsymbol{v}$. Thus $g$ is independent of $\boldsymbol{v}$, which implies that $\rho = \mathcal{R}^{-1} g$ is a radial function. In App. H.2 we give an analytic expression for $g$ for any isotropic distribution. In the special case of $\boldsymbol{X} \sim \mathcal{N}(\boldsymbol{0}, \boldsymbol{I})$, $g(\boldsymbol{v}, \cdot)$ decays monotonically with $b$, and thus, as discussed in Sec. 3.2, large step sizes will tend to increase the biases of all neurons. Additionally, we show in App. H.2 that for 2D data, $\rho$ is positive and strictly decreasing in $\|\boldsymbol{x}\|$, and it satisfies the asymptotics $\rho(\boldsymbol{x}) = O(\log(\|\boldsymbol{x}\|))$ as $\|\boldsymbol{x}\| \to 0$, and $\rho(\boldsymbol{x}) = O(\|\boldsymbol{x}\|^{-1})$ as $\|\boldsymbol{x}\| \to \infty$. Figure 3 visualizes $g$ and $\rho$ for two dimensional Gaussian data.

## 4  STABILITY LEADS TO DEPTH SEPARATION

Single hidden-layer neural networks are universal approximators, *i.e.*, they can approximate arbitrarily well any continuous function over compact sets (Pinkus, 1999). However, some of these approximations may correspond to unstable minima that are virtually unreachable by training via SGD. To understand what is the *effective* approximation power of neural networks, we need to identify the class of functions that have stable approximations. We have the following (see proof in App. I.1).

**Proposition 1.** *Let $X$ be the interior of the convex hull of the training points, and $f : X \to \mathbb{R}$ be any function. Suppose there exists a sequence of single hidden-layer ReLU networks $\{f_k\}$ with bounded stability norm that converges to $f$ in $L^1$ over $X$. Then $\|f\|_{\mathcal{R},g}$ is finite, and $\lim_{k \to \infty} \|f_k\|_{\mathcal{R},g} = \|f\|_{\mathcal{R},g}$.*

Let $\{f_k\}$ be a convergent sequence of stable solutions with a growing number of knots, *i.e.*, $\forall f_k \in \mathcal{F}_k : \|f_k\|_{\mathcal{R},g} \leq 1/\eta - 1/2$ (see Thm. 1). Then, by the proposition above we have that the limit function $f$ also satisfies this inequality. Therefore, the effective class of functions that can be approximated arbitrarily well by single hidden-layer ReLU networks includes only continuous functions $f$ that satisfy the stability condition $\|f\|_{\mathcal{R},g} \leq 1/\eta - 1/2$. As the step size decreases, more functions satisfy this condition, suggesting that more functions can be stably approximated by single hidden-layer ReLU networks. Surprisingly, there exists at least one continuous function $p$ that has $\|p\|_{\mathcal{R},g} = \infty$ and therefore does not satisfy the stability condition for *any* positive step size (see proof in App. I.2). Therefore from Prop. 1 and Thm. 1, this function cannot be approximated arbitrarily well by single hidden-layer ReLU networks trained with a non-vanishing step size.

**Proposition 2.** *Assume the input dimension $d \geq 2$, and let $p(\boldsymbol{x}) = \sigma(1 - \|\boldsymbol{x}\|_1)$. Suppose the support of $p$ is contained in the interior of the convex hull of the training points. Then $\|p\|_{\mathcal{R},g} = \infty$.*

Intriguingly, this function does have an implementation as a finite-width two hidden-layer network, $p(x) = \sigma(1 - \sum_{i=1}^d (\sigma(x_i) + \sigma(-x_i)))$, which is a stable solution for a fixed step size. Indeed, in App. I.3 we demonstrate that for an appropriate choice of $\eta$, GD is able to converge to this implementation. Thus, we have a depth separation result: the function $p$ cannot be approximated by stable minima of *one* hidden-layer networks trained with a non-vanishing step size, yet with *two* hidden-layers, GD can converge to this function with a fixed step size.

## 5    SHALLOW NETWORK APPROXIMATIONS OF SMOOTH FUNCTIONS

In Sec. 4 we showed that stable single-hidden layer ReLU networks are not universal approximators. In this section we give an approximation guarantee under smoothness assumptions. That is, we show that if a function is sufficiently smooth, then it can be approximated arbitrarily well using single hidden-layer networks that correspond to stable solutions of GD.

Let $W_w^{d+1,1}(\mathbb{R}^d)$ denote the weighted Sobolev space of all functions whose weak partial derivatives up to order $d+1$ are bounded in a weighted $L^1$-norm $\|\cdot\|_{1,w}$ with weight function $w(\boldsymbol{x}) := \mathcal{R}^*[1+|b|](\boldsymbol{x})$. Let $\|\cdot\|_{W_w^{d+1,1}(\mathbb{R}^d)}$ denote the corresponding Sobolev norm

$$\|f\|_{W_w^{d+1,1}(\mathbb{R}^d)} = \|f\|_{1,w} + \sum_{k=1}^{d+1} \sum_{|\beta|=k} \left\| \partial^\beta f \right\|_{1,w}, \tag{17}$$

where $\beta$ is a multi-index. For technical convenience, we restrict ourselves to odd input dimensions $d$ only. Our results use the "$\mathcal{R}$-norm" $\|\cdot\|_{\mathcal{R}}$ introduced by Ongie et al. (2020) (see Sec. 7 for details), and the stability norm $\|\cdot\|_{\mathcal{R},\hat{g}}$ with a different weight function $\hat{g}$ defined below (see proof in App. L).

**Proposition 3.** *Assume $d$ is odd and let $f \in W_w^{d+1,1}(\mathbb{R}^d)$. Then, there exists a sequence of single hidden-layer ReLU network functions $\{f_k\}$ such that $f_k \in \mathcal{F}_k$ converges to $f$ in $L^1$ over any compact subset $K \subset \mathbb{R}^d$, i.e., $\lim_{k \to \infty} \int_K |f_k(\boldsymbol{x}) - f(\boldsymbol{x})|\mathrm{d}\boldsymbol{x} = 0$, and satisfies the bounds $\|f_k\|_{\mathcal{R}} + \|f_k\|_{\mathcal{R},\hat{g}} \le c_{d,\hat{g}} \|f\|_{W_w^{d+1,1}(\mathbb{R}^d)}$ for all $k$, where*

$$\hat{g}(\boldsymbol{v}, b) = \mathbb{P}\left( \boldsymbol{X}^\top \boldsymbol{v} > b \right) \sqrt{\mathbb{E}\left[ \left( \boldsymbol{X}^\top \boldsymbol{v} - b \right)^2 \middle| \boldsymbol{X}^\top \boldsymbol{v} > b \right]} \sqrt{1 + \mathbb{E}\left[ \|\boldsymbol{X}\|^2 \middle| \boldsymbol{X}^\top \boldsymbol{v} > b \right]}, \tag{18}$$

*and $c_{d,\hat{g}}$ is a constant depending on $d$ and $\hat{g}$ but independent of $f$. Here $\boldsymbol{X}$ is drawn uniformly at random from the dataset.*

This proposition shows that for any $f \in W_w^{d+1,1}(\mathbb{R}^d)$ there exists a sequence of single hidden-layer ReLU network approximations $\{f_k\}$ for which $\{\|f_k\|_{\mathcal{R}}\}$ and $\{\|f_k\|_{\mathcal{R},\hat{g}}\}$ are bounded. To prove that these functions can have stable parameterizations for GD, we need to show that if both the stability norm and $\mathcal{R}$-norm are bounded (a function space property), then there exists a corresponding minimum with bounded sharpness[6] in parameter space. To this end, we derive an upper bound on the minimal sharpness of a solution $f$ among its different parameterizations in terms of the stability norm and the $\mathcal{R}$-norm (see proof in App. K).

**Lemma 2.** *Let $f \in \mathcal{F}_k$ be a solution for which the knots do not coincide with any training point. Then there exists an implementation $\boldsymbol{\theta}^*$ corresponding to $f$ such that*

$$\lambda_{\max}\left( \nabla^2 \mathcal{L}(\boldsymbol{\theta}^*) \right) \le 1 + 2\|f\|_{\mathcal{R},\hat{g}} + 4\left( \|f\|_{\mathcal{R}} + \inf_{\boldsymbol{x} \in \mathbb{R}^d} \|\nabla f(\boldsymbol{x})\| \right) \sqrt{\lambda_{\max}(\boldsymbol{\Sigma_X})} \sqrt{1 + \mathbb{E}\left[ \|\boldsymbol{X}\|^2 \right]}. \tag{19}$$

*Here $\boldsymbol{X}$ is drawn uniformly at random from the dataset, and $\boldsymbol{\Sigma_X}$ is the covariance matrix of $\boldsymbol{X}$.*

Combining Prop. 3 and Lemma 2 we get that any $f \in W_w^{d+1,1}(\mathbb{R}^d)$ can be approximated arbitrarily well by a sequence of stable solutions for GD with a *fixed* step size $\eta$.

**Theorem 2.** *Suppose the input dimension $d$ is odd, and let $f \in W_w^{d+1,1}(\mathbb{R}^d)$. Then, there exist $\eta > 0$ and a sequence of single hidden-layer ReLU network functions $\{f_k\}$ such that $f_k \in \mathcal{F}_k$ converges to $f$ in $L^1$ over any compact subset $K \subset \mathbb{R}^d$, and every $f_k$ is stable for GD with step size $\eta$.*

This theorem state that any sufficiently smooth function can be stably approximated in the limit of infinitely many neurons. We can also use Lemma 2 to guarantee the stability of solutions in the finite case. Since $\lambda_{\max} \le 2/\eta$ is a sufficient condition for stability in GD, we have the following.

**Theorem 3.** *Let $f \in \mathcal{F}_k$ be a solution for which the knots do not coincide with any training point. If*

$$\|f\|_{\mathcal{R},\hat{g}} + 2\left( \|f\|_{\mathcal{R}} + \inf_{\boldsymbol{x} \in \mathbb{R}^d} \|\nabla f(\boldsymbol{x})\| \right) \sqrt{\lambda_{\max}(\boldsymbol{\Sigma_X})} \sqrt{1 + \mathbb{E}\left[ \|\boldsymbol{X}\|^2 \right]} \le \frac{1}{\eta} - \frac{1}{2}, \tag{20}$$

*then $f$ is a stable solution for GD with step size $\eta$.*

Theorem 3 complements Thm. 1, as it gives a sufficient condition for stability in function space.

---

[6]Note that for GD, $\eta < 2/\lambda_{\max}$ is a necessary and sufficient condition for linear stability.

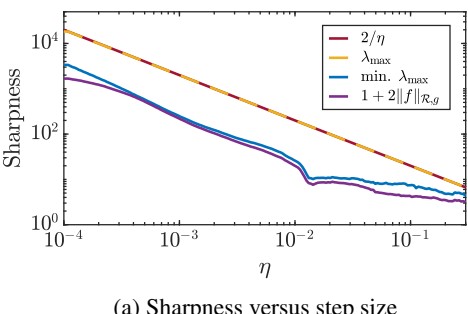
(a) Sharpness versus step size

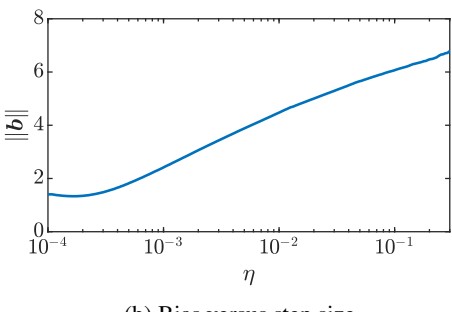
(b) Bias versus step size

Figure 4: **Validating the bounds on synthetic data.** We trained a two-layer ReLU network on a regression task with synthetic data using GD (see Sec. 6). Panel (a) depicts the sharpness of the minima to which GD converged, as a function of the step size $\eta$. As $\eta$ increases, the minima get flatter in parameter space (yellow curve), which translates to smoother predictors in function space (purple curve). Panel (b) shows the norm of the bias vector $\boldsymbol{b}$ as a function of the step size. Here we see that the bias vector grows with the step size, as the predictor function gets smoother.

## 6 EXPERIMENTS

We now demonstrate our theoretical results. We start with a regression task on synthetic data. Here, we drew $n = 100$ pairs $(\boldsymbol{x}_j, y_j)$ in $\mathbb{R}^{20} \times \mathbb{R}$ from the standard normal distribution to serve as our training set. We fit a single hidden-layer ReLU network with $k = 40$ neurons to the data using GD with various step sizes (runs were stopped when the loss dropped bellow $10^{-8}$). For each minimum $\boldsymbol{\theta}^*$ to which GD converged, we computed the loss' sharpness, $\lambda_{\max}(\nabla^2 \mathcal{L}(\boldsymbol{\theta}^*))$, and our lower bound on the sharpness, $1 + 2\|f\|_{\mathcal{R},g}$ (Lemma 3 in the appendix). Additionally, we numerically determined the sharpness of the flattest implementation for every solution. Figure 4(a) depicts the results for this experiment. The red line marks the border of the stable region, which is $2/\eta$. Namely, (S)GD cannot stably converge to a minimum whose sharpness is above this line. The dashed yellow line shows the sharpness of the minima to which GD converged in practice. As can be seen, here GD converged at the edge of stability (the two lines coincide), a phenomenon discussed in (Cohen et al., 2020). The blue curve is the sharpness of the flattest implementation of each solution (see App. A), while the the purple curve is our lower bound. We see that our bound is quite tight (blue vs. purple). Furthermore, as the step size increases the minima get flatter in parameter space (yellow curve), which translates to smoother predictors in function space (purple curve). Additionally, we see from Fig. 4(b) that the norm of the bias vector $\boldsymbol{b}$ increases with the step size, as our theory predicts (Sec. 3.2).

Next, we present an experiment with binary classification on MNIST (LeCun, 1998) using SGD. In this experiment we used $n = 512$ samples from two MNIST classes, '0' and '1'. The classes were labeled as $y = 1$ and $y = -1$, respectively. For the validation set we used 4000 images from the remaining samples in each class. We trained a single hidden-layer ReLU network with $k = 200$ neurons using SGD with batch size $B = 16$, and the quadratic loss. To perform classification at inference time, we thresholded the net's output at 0. We ran SGD until the loss dropped below $10^{-8}$ for 2000 consecutive epochs. Figure 5(a) shows the same indices as in the previous experiment. Here we see again that as the step size increases, the minima get flatter in parameter space (yellow curve), which translates to smoother predictors in function space (purple curve). Figure 5(b) shows the classification accuracy on the validation set, where we see that the network generalizes better as the step size increases, as past work showed, *e.g.*, (Keskar et al., 2017). More experiments in App. N.

## 7 RELATED WORK

Dynamical stability analysis was applied to neural network training in several works. In particular, Nar & Sastry (2018) analyzed Lyapunov stability of one hidden-layer ReLU networks *without bias*. They proved a bound on the network's output which depends on the step size and training data and implies that the network's output should be smaller for training samples with larger magnitude. Mulayoff & Michaeli (2020) characterized the flattest minima for linear nets and showed that these minima have unique properties. Yet, in their setting all minima implement the same input-output

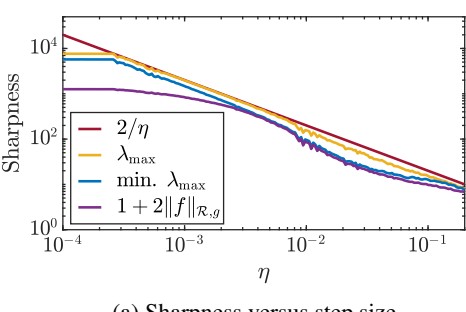
(a) Sharpness versus step size

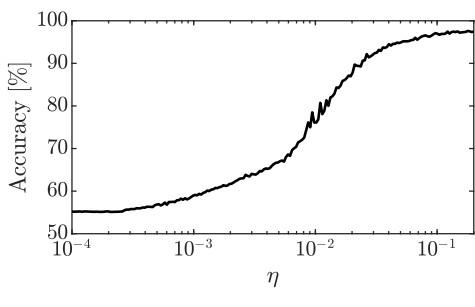
(b) Val. accuracy vs. step size

Figure 5: **Validating the bounds on MNIST.** We trained a single hidden-layer ReLU network for binary classification on two classes from MNIST using SGD (see Sec. 6). Panel (a) depicts the sharpness versus the step size $\eta$. Here as $\eta$ increases, the minima get flatter in parameter space (yellow curve), which translates to smoother predictors in function space (purple curve). Panel (b) shows the performance on the validation set. Here the trained model generalizes better as the step size increases.

function. Thus, their results only show that SGD is biased toward certain *implementations* of the same function, whereas our result shows that SGD is biased toward certain *functions*.

Wu et al. (2018) proved a sufficient condition for dynamical stability of SGD in expectation using second moment. Ma & Ying (2021) extended their result by showing a necessary and sufficient condition for dynamical stability in higher moments. In addition, the authors combined this condition with the multiplicative structure of neural nets to prove an upper bound on the Sobolev seminorm of the model's input-output function at stable interpolating solutions. Their upper bound extends to deep nets, yet it depends on the norm of the first layer of the network, which in general can be large.

Mulayoff et al. (2021) characterized the stable solutions of SGD for *univariate* single hidden-layer ReLU networks with the square loss. Our Thm. 1 is the natural extension of (Mulayoff et al., 2021, Thm. 1) to the multivariate case. To prove it, we combine the proof technique of lower bounding the top eigenvalue of the Hessian, used by Mulayoff et al. (2021), with the Radon transforms analysis used by Ongie et al. (2020). Combining these techniques is not *a priori* trivial, since Radon transform was not used before for Hessian analysis. Also, it required several subtle steps that are not encountered in the univariate setting nor in (Ongie et al., 2020) (*e.g.*, working with the inverse of the dual Radon transform to obtain the primal space representation).

Ongie et al. (2020) studied the space of functions realizable as infinite-width single hidden-layer ReLU nets with bounded weights norm. Their settings assumes *explicit* regularization, *i.e.*, min-norm solution, whereas here we derived our results for SGD *without* regularization, via implicit bias. On the technical level, they introduced the "$\mathcal{R}$-norm" $\|\cdot\|_{\mathcal{R}}$ that is closely related to the stability norm. Particularly, $\|\cdot\|_{\mathcal{R}} = \|\cdot\|_{\mathcal{R},g}$, for $g = \mathbf{1}$ the constant 1 function. They proved similar results of depth separation and approximation guarantees, shown here in Secs. 4-5. More related work in App. B.

## 8 CONCLUSION

Large step sizes are often used to improve generalization (Li et al., 2019). This work suggests an explanation to this practice. Specifically, we showed that large step sizes lead to smaller stability norm and thus can bias towards smooth predictors in shallow multivariate ReLU networks. We find the smoothness measure depends on the data via specific functions $g$ or $\rho$, and exemplify their properties. Moreover, we studied the approximation power of ReLU networks that correspond to stable solutions. Although shallow networks are universal approximators, we proved that stable solutions of these networks are not. Namely, there is a function that cannot be well-approximated by stable single hidden-layer ReLU networks trained with a non-vanishing step size. Yet we showed that the same function can be realized as a stable two hidden-layer network, leading to a depth separation result. This result can explain the success of deep models over shallow ones. Finally, we gave approximation guarantees for stable shallow ReLU networks. In particular, we proved that any Sobolev function can be approximated arbitrarily well using GD with single hidden-layer ReLU networks.

ACKNOWLEDGMENTS

The research of RM was supported by the Planning and Budgeting Committee of the Israeli Council for Higher Education, and by the Andrew and Erna Finci Viterbi Graduate Fellowship. GO was supported by NSF CRII award CCF-2153371. The research of DS was funded by the European Union (ERC, A-B-C-Deep, 101039436). Views and opinions expressed are however those of the author only and do not necessarily reflect those of the European Union or the European Research Council Executive Agency (ERCEA). Neither the European Union nor the granting authority can be held responsible for them. DS also acknowledges the support of Schmidt Career Advancement Chair in AI. TM was supported by grant 2318/22 from the Israel Science Foundation and by the Ollendorff Center of the Viterbi Faculty of Electrical and Computer Engineering at the Technion.

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

## A  ADDITIONAL DISCUSSION

**The independence of Lemma 1 and Theorem 1 on the batch size $B$.**  Theorem 1 relies on Lemma 1 (see App. E), which was proved in (Mulayoff et al., 2021). This lemma states that if a minimum $\boldsymbol{\theta}^*$ is linearly stable for SGD with batch-size $B$, then the Hessian $\boldsymbol{H}$ of the loss at $\boldsymbol{\theta}^*$ must satisfy $\lambda_{\max}(\boldsymbol{H}) \leq 2/\eta$, where $\eta$ is the step-size. Importantly, this necessary condition holds true for any batch size $B$, and thus Theorem 1 is independent of $B$. We note, however, that the precise stability threshold of SGD might depend on $B$, yet the important points to notice are: (1) Here all we need is a necessary condition, and Lemma 1 provides a simple bound that holds for any $B$. (2) Empirical evidence shows that there is not much room for improvement upon this batch-size-independent bound in real-world settings. Specifically, for practical batch sizes, the gap between $2/\eta$ and the stability threshold of SGD is often very small (see Fig. 5 in our paper and Figures 2-3 in (Gilmer et al., 2022)).

The proof of Lemma 1 is actually quite short and easy to follow (see (Mulayoff et al., 2021, App. II)). The idea is that if $\boldsymbol{\theta}^*$ is an $\varepsilon$ linearly stable minimum, then by definition we have

$$\limsup_{t \to \infty} \mathbb{E}[\|\boldsymbol{\theta}_t - \boldsymbol{\theta}^*\|] \leq \varepsilon, \tag{21}$$

where $\{\boldsymbol{\theta}_t\}_{t=0}^{\infty}$ are governed by the linearized stochastic dynamics given in Eq. (5). Using Jensen's inequality, for all $t > 0$ we get $\|\mathbb{E}[\boldsymbol{\theta}_t] - \boldsymbol{\theta}^*\| \leq \mathbb{E}[\|\boldsymbol{\theta}_t - \boldsymbol{\theta}^*\|]$. Thus,

$$\limsup_{t \to \infty} \|\mathbb{E}[\boldsymbol{\theta}_t] - \boldsymbol{\theta}^*\| \leq \limsup_{t \to \infty} \mathbb{E}[\|\boldsymbol{\theta}_t - \boldsymbol{\theta}^*\|] \leq \varepsilon. \tag{22}$$

Note that under the linearized dynamics, $\{\mathbb{E}[\boldsymbol{\theta}_t]\}_{t=0}^{\infty}$ are precisely GD steps. Therefore, we have that if $\boldsymbol{\theta}^*$ is linearly stable for SGD, then it must be linearly stable also for GD. Now, a well-known fact is that $\boldsymbol{\theta}^*$ is linearly stable for GD if and only if $\lambda_{\max}(\boldsymbol{H}) \leq 2/\eta$. This is how we get the necessary condition in Lemma 1, which does not depend on the batch size.

**The independence of Lemma 1 and Theorem 1 on $\varepsilon$.**  Lemma 1 states that a necessary condition for a twice differentiable minimum to be $\varepsilon$ linearly stable is $\lambda_{\max}\left(\nabla^2 \mathcal{L}(\boldsymbol{\theta}^*)\right) \leq 2/\eta$. That is, the condition does not depend on $\varepsilon$ which might seem not intuitive. However, the reason Lemma 1 does not depend on $\varepsilon$ is because it refers to linear stability, opposed to non-linear dynamical stability. In linear stability for twice-differentiable minima, all we care about is the second-order Taylor approximation of the loss at the minimum. In see previous paragraph we explained that Lemma 1 gives a necessary condition through reduction to GD. Now, when applying GD on a quadratic loss (with a PSD matrix), for any $\varepsilon > 0$ only one of two things can happen:

1. Either $\exists \boldsymbol{\theta}_0 \in \mathcal{B}_\varepsilon(\boldsymbol{\theta}^*) : \ \limsup_{t \to \infty} \|\boldsymbol{\theta}_t - \boldsymbol{\theta}^*\| = +\infty$ (unstable for any $\varepsilon > 0$),

2. or $\forall \boldsymbol{\theta}_0 \in \mathcal{B}_\varepsilon(\boldsymbol{\theta}^*) : \quad \limsup_{t \to \infty} \|\boldsymbol{\theta}_t - \boldsymbol{\theta}^*\| \leq \|\boldsymbol{\theta}_0 - \boldsymbol{\theta}^*\| \leq \varepsilon$ (stable for any $\varepsilon > 0$).

In any outcome, the result does not depend on $\varepsilon$, and therefore $\varepsilon$ does not appear in the result of Lemma 1. Note that for non-differentiable minima, $\varepsilon$ does affect linear stability in GD, however Lemma 1 only refers to twice-differentiable minima.

Theorem 1 is based on Lemma 1, and therefore does not depend on $\varepsilon$. Yet, beyond this technical reasoning, it is important to note that here we consider *interpolating solutions*. For those solutions, the global minimum of the loss is also a global minimum w.r.t. each data sample $(\boldsymbol{x}_j, y_j)$ separately. Therefore, despite the stochasticity of SGD, every step points towards a global minimum. This implies that if the stability criterion is satisfied, then SGD converges to the minimum ($\limsup_{t \to \infty} \mathbb{E}[\|\boldsymbol{\theta}_t - \boldsymbol{\theta}^*\|] = 0$) and if it is not satisfied, then SGD repels from the minimum ($\limsup_{t \to \infty} \mathbb{E}[\|\boldsymbol{\theta}_t - \boldsymbol{\theta}^*\|] = \infty$ for the linearized dynamics). This is also seen in simulations where models are overfit to training data using SGD, *e.g.*, (Ma et al., 2018b). Particularly, in our simulations the loss always converged to 0 when it converged (we arbitrarily decided to stop each run when the loss dropped below $10^{-8}$). In the general case of non-interpolating solutions, the expected final distance to the minimum in mini-batch SGD ($\limsup_{t \to \infty} \mathbb{E}[\|\boldsymbol{\theta}_t - \boldsymbol{\theta}^*\|]$) can be a strictly positive finite number, and therefore in those cases $\varepsilon$ does play a role.

**Large step size training and warmup.** While Theorem 1 applies to any positive step size, it is most interesting when considering large step sizes. High learning rates are standard practice, as they are associated with good generalization (Li et al., 2019). However, there are cases, *e.g.*, large initialization, in which high learning rate might cause the training to diverge. In these cases, a learning rate warmup is applied, enabling training with large step sizes.

**Learning rate decay.** Practitioners often work with learning rate schedule, which typically reduces the step size toward the end of training. In this scenario, $2/\eta$ can be quite high at the end, making Theorem 1 loose. Here, empirical evidence shows that when reducing the step size at a late stage, the sharpness of the obtained minimum is often still controlled by the initial step size, *e.g.*, Figs. 1 and 3 in (Gilmer et al., 2022). Moreover, although learning rate decay is a popular technique, there are other popular training schemes in which the learning rate is not reduced, *e.g.*, (Smith et al., 2018). Lastly, as for the depth separation results (Sec. 4) and the approximation results (Sec. 5), they apply for *any* fixed positive step size. In other words, the learning rate decay does not affect these results.

**Initialization independent results.** Our results are independent of the initialization. In the past it was shown that initialization can have large effect on which minimum GD converges under certain conditions. However, this do not contradict our results, as explained below.

For very small step sizes, the GD trajectory follows that of gradient flow (GF). Under certain conditions, *e.g.*, infinite width or vanishing initialization, it was shown that the network does not change much on the evolution trajectory of GF. In this case, the initialization dominates the properties of the obtained solution. This is known as kernel regime or Neural Tangent Kernel (NTK) regime (Jacot et al., 2018; Chizat et al., 2019). However, for practical step sizes and standard initialization, recent work (Cohen et al., 2020) showed that GD typically deviates from the GF trajectory, while entering the Edge of Stability regime. This occurs when the stability threshold is achieved during training, *i.e.*, $\lambda_{\max}(\nabla^2 \mathcal{L}(\boldsymbol{\theta}_t)) \geq 2/\eta$ for some $t > 0$. In this case, GD converges to a different minimum than GF, *i.e.*, GD escapes the NTK regime. Similar behavior was shown also for SGD (Gilmer et al., 2022).

**Theorem 1 Proof idea.** Theorem 1 is a result of two properties of twice-differentiable minima. First, we show in Lemma 3 in the appendix that these minima satisfy $\lambda_{\max}(\nabla_{\boldsymbol{\theta}}^2 \mathcal{L}) \geq 1 + 2\|f\|_{\mathcal{R},g}$. Second, we know from Lemma 1 that stable minima satisfy $\lambda_{\max}(\nabla_{\boldsymbol{\theta}}^2 \mathcal{L}) \leq 2/\eta$. Together, these properties imply that $2/\eta \geq 1 + 2\|f\|_{\mathcal{R},g}$, from which we get the result of the theorem, $\|f\|_{\mathcal{R},g} \leq 1/\eta - 1/2$. Note that twice-differentiable minima correspond to functions whose knots do not coincide with any training point. Although we prove our result only for such functions, we observed that in practice the condition $\lambda_{\max}(\nabla_{\boldsymbol{\theta}}^2 \mathcal{L}) \leq 2/\eta$ is always met around minima to which SGD converges (see Sec. 6 and the next paragraph). For full derivation of the theorem see App. E.

**Theorem 1 assumption and minima that are not twice-differentiable.** Theorem 1 assumes that the knots of $f$ do not coincide with any training point. This assumption is done for technical simplicity, that is to ensure the minimum is twice-differentiable. In practice, we observed that usually only a small fraction of the knots coincide with training points. For example, in our MNIST experiment we had only 30 training points coinciding with knots of $f$ (out of $n = 512$ training points and $k = 200$ neurons). Note that the twice-differentiability assumption in Theorem 1 is required for Lemma 1 to hold as Theorem 1 invokes Lemma 1. However, we observed that in practical settings, Lemma 1 still applies in settings where the assumption is violated. This can be appreciated by the red curve upper bounding the orange curve in Figs. 4(a) and 5(a). Namely, despite the fact that the minima are not always twice-differentiable in our experiments, the stability criterion for SGD is still observed to be upper bounded by $2/\eta$.

It is possible to extend the analysis to minima that are not twice-differentiable by using the same method as in (Mulayoff et al., 2021). However, this makes the analysis much more complex.

**The meaning of** $\min \lambda_{\max}$ **in figures 4(a) and 5(a).** The curve $\min \lambda_{\max}$ shows the sharpness of the flattest implementation of each solution $f$. In more detail, as discussed earlier, Theorem 1 is a result of two properties of twice-differentiable minima. On the one hand, we know from Lemma 1 that stable minima satisfy $\lambda_{\max}(\nabla_{\boldsymbol{\theta}}^2 \mathcal{L}) \leq 2/\eta$. On the other hand, Lemma 3 in App. E asserts that these minima satisfy $\lambda_{\max}(\nabla_{\boldsymbol{\theta}}^2 \mathcal{L}) \geq 1 + 2\|f\|_{\mathcal{R},g}$. Note that each function $f \in \mathcal{F}_k$ has multiple

implementations, *i.e.*, different minima in parameter space which all correspond to $f$. These minima can have different sharpness. Here, we can look at the best implementation, *i.e.*, a solution to $\min \lambda_{\max}$, where the minimum is taken over all loss' minima $\{\boldsymbol{\theta}\}$ that implement $f$. Overall, given a minimum $\boldsymbol{\theta}$ with a corresponding function $f$, we have

$$1 + 2\|f\|_{\mathcal{R},g} \le \min \lambda_{\max}(\nabla_{\boldsymbol{\theta}}^2 \mathcal{L}) \le \lambda_{\max}(\nabla_{\boldsymbol{\theta}}^2 \mathcal{L}) \le 2/\eta. \tag{23}$$

These inequalities give us the result of Theorem 1, $1 + 2\|f\|_{\mathcal{R},g} \le 1/\eta - 1/2$. To understand the tightness of each part of our analysis, we added to the plots $\lambda_{\max}$ and $\min \lambda_{\max}$. In both figures, $\lambda_{\max}(\nabla_{\boldsymbol{\theta}}^2 \mathcal{L})$ equals or just below $2/\eta$, a phenomenon known as edge of stability (Cohen et al., 2020). Additionally, $\min \lambda_{\max}(\nabla_{\boldsymbol{\theta}}^2 \mathcal{L})$ is close to $1 + 2\|f\|_{\mathcal{R},g}$ in these experiments. Yet, Figure 4 shows that $\lambda_{\max}(\nabla_{\boldsymbol{\theta}}^2 \mathcal{L})$ can be quite larger than $\min \lambda_{\max}(\nabla_{\boldsymbol{\theta}}^2 \mathcal{L})$, meaning that there exists a far flatter minimum that implements the same function. This fact was used by Dinh et al. (2017) to show that sharp minima can generalize.

## B  ADDITIONAL RELATED WORK

**Implicit bias.**  A long line of works studied the implicit bias of the training procedure in an attempt to better understand generalization in overparameterized models. For the classification setting, in the case of linear prediction function, linearly separable data, and exponentially tailed loss functions (*e.g.*, logistic and exponential), Soudry et al. (2018) showed that GD converges in the direction of the SVM solution. This result was later extended to linear fully connected and convolutional neural networks (Gunasekar et al., 2018b; Ji & Telgarsky, 2019a), more loss functions (Nacson et al., 2019b; Ji & Telgarsky, 2021), SGD optimization algorithm (Nacson et al., 2019c), other generic optimization methods (Gunasekar et al., 2018a), non-separable data (Ji & Telgarsky, 2019b), and homogeneous prediction functions (Nacson et al., 2019a; Lyu & Li, 2020; Ji et al., 2020). However, all those results do not depend on the step size, except for the requirement that it be sufficiently small.

Another line of works studied the implicit bias in the context of linear models with quadratic loss such as matrix factorization (Gunasekar et al., 2017; Li et al., 2018; Arora et al., 2018; 2019; Belabbas, 2020; Eftekhari & Zygalakis, 2021; Gidel et al., 2019; Ma et al., 2018a; Woodworth et al., 2020; Azulay et al., 2021). However, all of these works relied on either small or infinitesimal step size (*i.e.*, gradient flow). Thus, they do not capture how the step size affects the implicit bias. Moreover, they assumed a manifold property (Azulay et al., 2021). As pointed out by Razin & Cohen (2020) and Vardi & Shamir (2021), these assumptions do not always apply. In contrast, our result is based on a stability condition of SGD, which depends on the step size and does not require the manifold assumption.

**How the step size affects the implicit bias.**  To investigate the implicit bias of the step size, Barrett & Dherin (2021) and Smith et al. (2021) suggested using a modified loss. Under this modified loss, gradient flow approximates the trajectory of (S)GD on the original loss. However, the step size should be sufficiently small for the approximation to hold true. Moreover, the induced regularization term this method yields is expressed in terms of the model's parameters. Additionally, this term increases linearly with the step size and vanishes at any stationary point.

**Radon transform analysis of shallow networks.**  Radon transform analysis has previously been used in studies of the approximation capabilities of single hidden-layer neural networks with bounded activation functions (Carroll & Dickinson, 1989; Ito, 1991), and more general activation functions in the ridgelet framework (Candès & Donoho, 1999; Candès, 1999). More recently, Sonoda & Murata (2017) used ridgelet transform analysis to study the approximation properties of two-layer neural networks with unbounded activation functions, including the ReLU.

Parts of this work extend results by Ongie et al. (2020), which defined a similar Radon-domain seminorm (the "$\mathcal{R}$-norm") to determine the space of functions realizable as an infinite-width single hidden-layer ReLU networks with square-summable weights. Parhi & Nowak (2021) proved a representer theorem for single hidden-layer ReLU networks using the $\mathcal{R}$-norm. Finally, an $L^2$ version of the $\mathcal{R}$-norm is used by Jin & Montúfar (2020) to describe the function space implicit bias of training a single hidden-layer ReLU network using gradient descent in the neural tangent kernel regime.

## C  THE RADON TRANSFORM

For a function $f : \mathbb{R}^d \to \mathbb{R}$, the $d$-dimensional Radon transform $\mathcal{R}f$ is the collection of all integrals of $f$ over $(d-1)$-dimensional affine hyperplanes in $\mathbb{R}^d$. Every hyperplane can be parametrized by a pair $(\boldsymbol{v}, b) \in \mathbb{S}^{d-1} \times \mathbb{R}$, where $\boldsymbol{v}$ is a unit normal to the hyperplane and $b \in \mathbb{R}$ is its distance from the origin. Therefore, the Radon transform $\mathcal{R}f$ is the function over $(\boldsymbol{v}, b) \in \mathbb{S}^{d-1} \times \mathbb{R}$ given by

$$\mathcal{R}f(\boldsymbol{v}, b) \triangleq \int_{\boldsymbol{v}^\top \boldsymbol{x} = b} f(\boldsymbol{x}) \mathrm{d}s(\boldsymbol{x}), \tag{24}$$

where $\mathrm{d}s(\boldsymbol{x})$ represents integration with respect to the $(d-1)$-dimensional surface measure on the hyperplane $\boldsymbol{v}^\top \boldsymbol{x} = b$. Note that the Radon transform is an even function, *i.e.*, $\mathcal{R}f(\boldsymbol{v}, b) = \mathcal{R}f(-\boldsymbol{v}, -b)$, since $(\boldsymbol{v}, b)$ and $(-\boldsymbol{v}, -b)$ describe the same hyperplane.

The dual Radon transform $\mathcal{R}^*$ maps functions defined on $\mathbb{S}^{d-1} \times \mathbb{R}$ to functions on $\mathbb{R}^d$ by

$$\mathcal{R}^*\varphi(\boldsymbol{x}) \triangleq \int_{\mathbb{S}^{d-1}} \varphi\left(\boldsymbol{v}, \boldsymbol{v}^\top \boldsymbol{x}\right) \mathrm{d}s(\boldsymbol{v}) \tag{25}$$

for all $\boldsymbol{x} \in \mathbb{R}^d$, where $\mathrm{d}s(\boldsymbol{v})$ represents integration with respect to the $(d-1)$-dimensional surface measure on the unit sphere $\mathbb{S}^{d-1}$.

The Radon transform and its dual are invertible over spaces of smooth functions via the inversion formulas:

$$\mathcal{R}^{-1} = \gamma_d (-\Delta)^{\frac{d-1}{2}} \mathcal{R}^*, \tag{26}$$

$$(\mathcal{R}^*)^{-1} = \gamma_d \mathcal{R}(-\Delta)^{\frac{d-1}{2}}, \tag{27}$$

where the fractional Laplacian operator $(-\Delta)^{\frac{d-1}{2}}$ is defined by application of a ramp function in Fourier domain (*i.e.*, multiplication by $\|\boldsymbol{\omega}\|^{d-1}$ in Fourier domain), and

$$\gamma_d = \frac{1}{2(2\pi)^{d-1}} \tag{28}$$

is a dimension dependent constant.

These transforms may be extended to spaces of distributions (*e.g.*, Dirac deltas) in a standard way (Ludwig, 1966; Helgason, 1999), which we summarize in App. D. Important for this work is the distributional dual inverse Radon transform $(\mathcal{R}^*)^{-1}$, which maps a distribution defined over Euclidean space $\mathbb{R}^d$ to a distribution in Radon domain $\mathbb{S}^{d-1} \times \mathbb{R}$.

## D  DISTRIBUTIONAL FRAMEWORK

Let $f : \mathbb{R}^d \to \mathbb{R}$ be any locally integrable function. Then its Laplacian $\Delta f$ can be interpreted as a tempered distribution, meaning that $\Delta f$ is defined via the duality pairing

$$\langle \Delta f, \varphi \rangle \triangleq \langle f, \Delta\varphi \rangle_{\mathbb{R}^d} = \int_{\mathbb{R}^d} f(\boldsymbol{x}) \Delta\varphi(\boldsymbol{x}) \mathrm{d}\boldsymbol{x}, \tag{29}$$

where $\varphi$ is any Schwartz test function on $\mathbb{R}^d$, *i.e.*, a smooth function such that the function and its partial derivatives of all orders have sufficiently fast decay at infinity; denote this space of functions by $\mathcal{S}(\mathbb{R}^d)$. For example, if $f$ consists of a single ReLU unit, *i.e.*, $f(\boldsymbol{x}) = \sigma(\boldsymbol{v}^\top \boldsymbol{x} - b)$ such that $\|\boldsymbol{v}\| = 1$, then it is easy to show $\Delta f = \delta(\boldsymbol{v}^\top \boldsymbol{x} - b)$, meaning $\langle \Delta f, \varphi \rangle = \int_{\{x : \boldsymbol{v}^\top \boldsymbol{x} = b\}} \varphi(\boldsymbol{x}) \mathrm{d}\boldsymbol{x} = \mathcal{R}\varphi(\boldsymbol{v}, b)$. In other words, $\Delta f$ is the distribution given by evaluation of the Radon transform of a test function at the point $(\boldsymbol{v}, b) \in \mathbb{S}^{d-1} \times \mathbb{R}$.

Next, we describe how to understand the operator $(\mathcal{R}^*)^{-1}$ in a distributional sense. Let $\mathcal{S}_H(\mathbb{S}^{d-1} \times \mathbb{R})$ denote the image of Schwartz functions $\mathcal{S}(\mathbb{R}^d)$ under the classical Radon transform $\mathcal{R}$. The space $\mathcal{S}_H(\mathbb{S}^{d-1} \times \mathbb{R})$ is characterized in (Ludwig, 1966; Helgason, 1999); it is the space of all even Schwartz functions defined on $\mathbb{S}^{d-1} \times \mathbb{R}$ that additionally satisfy some moment conditions[7]. It is also shown

---

[7] Specifically, for all positive integers $k$, the function $\boldsymbol{v} \to \int_{b \in \mathbb{R}} \phi(\boldsymbol{v}, b) b^k \mathrm{d}b$ needs to be a homogeneous polynomial in $\boldsymbol{v}$ of degree $k$.

by Ludwig (1966) that the classical inverse Radon transform $\mathcal{R}^{-1}$ is a linear homeomorphism of $\mathcal{S}_H(\mathbb{S}^{d-1} \times \mathbb{R})$ onto $\mathcal{S}(\mathbb{R}^d)$. Therefore, we may define its distributional transpose $(\mathcal{R}^{-1})^* = (\mathcal{R}^*)^{-1}$ applied to any tempered distribution $h$ by

$$\langle (\mathcal{R}^*)^{-1} h, \phi \rangle = \langle h, \mathcal{R}^{-1} \phi \rangle \tag{30}$$

for all $\phi \in \mathcal{S}_H(\mathbb{S}^{d-1} \times \mathbb{R})$. Note that $(\mathcal{R}^*)^{-1} h$ is a distribution belonging to $\mathcal{S}'_H(\mathbb{S}^{d-1} \times \mathbb{R})$, the topological dual of $\mathcal{S}_H(\mathbb{S}^{d-1} \times \mathbb{R})$.

Returning to the example where $f$ is a single ReLU unit, *i.e.*, $f(\boldsymbol{x}) = \sigma(\boldsymbol{v}^\top \boldsymbol{x} - b)$ with $\|\boldsymbol{v}\| = 1$, then for any test function $\phi \in \mathcal{S}_H(\mathbb{S}^{d-1} \times \mathbb{R})$ we have

$$\langle (\mathcal{R}^*)^{-1} \Delta f, \phi \rangle = \langle \Delta f, \mathcal{R}^{-1} \phi \rangle = [\mathcal{R} \mathcal{R}^{-1} \phi](\boldsymbol{v}, b) = \phi(\boldsymbol{v}, b). \tag{31}$$

This shows $(\mathcal{R}^*)^{-1} \Delta f = \delta_{(\boldsymbol{v}, b)}$, *i.e.*, a Dirac delta centered at $(\boldsymbol{v}, b)$. If $f(\boldsymbol{x}) = \sum_{i=1}^k a_i \sigma(\boldsymbol{v}_i^\top \boldsymbol{x} - b_i) + c$ is any single hidden-layer ReLU network such that $\|\boldsymbol{v}_i\| = 1$ for all $i = 1, ..., k$, then by linearity we have

$$(\mathcal{R}^*)^{-1} \Delta f = \sum_{i=1}^k a_i \delta_{(\boldsymbol{v}_i, b_i)}. \tag{32}$$

Finally, we may define the total variation $\|\cdot\|_{\mathrm{TV}}$ for any distribution $\alpha \in \mathcal{S}'_H(\mathbb{S}^{d-1} \times \mathbb{R})$ by

$$\|\alpha\|_{\mathrm{TV}} \triangleq \sup_{\phi \in \mathcal{S}_H(\mathbb{S}^{d-1} \times \mathbb{R})} |\langle \alpha, \phi \rangle|. \tag{33}$$

If $\|\alpha\|_{\mathrm{TV}}$ is finite, then $\alpha$ is a distribution of order-0. In this case, since $\mathcal{S}_H(\mathbb{S}^{d-1} \times \mathbb{R})$ is dense in the space of even and continuous functions on $\mathbb{S}^{d-1} \times \mathbb{R}$ that vanish at infinity, $\alpha$ can be extended uniquely to an even signed measure on $\mathbb{S}^{d-1} \times \mathbb{R}$, and $\|\alpha\|_{\mathrm{TV}}$ is equal to the total variation norm of $\alpha$.

# E    PROOF OF THEOREM 1

In the proof of the theorem we use the following lemma (for the proof of this lemma see Appendix F).

**Lemma 3** (Top eigenvalue lower bound). *Let $f \in \mathcal{F}_k$ be a twice-differentiable minimizer of the loss function, then*

$$\lambda_{\max}\left(\nabla_{\boldsymbol{\theta}}^2 \mathcal{L}\right) \geq 1 + 2 \|f\|_{\mathcal{R}, g}, \tag{34}$$

*where $\|\cdot\|_{\mathcal{R}, g}$ denotes the stability norm.*

Let $f \in \mathcal{F}_k$ be a stable solution of the loss function. Then, according to Definition 3, there exists a linearly stable minimum point $\boldsymbol{\theta} \in \mathbb{R}^{(d+2)k+1}$ such that the network at this minimum implements $f$. Due to the fact that the knots of $f$ do not contain any training point, we have that $\boldsymbol{\theta}$ is a twice-differentiable minimum. From Lemma 1, since $\boldsymbol{\theta}$ is a twice differentiable stable minimum then

$$\lambda_{\max}\left(\nabla_{\boldsymbol{\theta}}^2 \mathcal{L}\right) \leq \frac{2}{\eta}. \tag{35}$$

On the other hand, from Lemma 3 we have that

$$\lambda_{\max}\left(\nabla_{\boldsymbol{\theta}}^2 \mathcal{L}\right) \geq 1 + 2 \|f\|_{\mathcal{R}, g}. \tag{36}$$

Using (35) and (36) we get

$$\|f\|_{\mathcal{R}, g} \leq \frac{1}{\eta} - \frac{1}{2}. \tag{37}$$

# F    PROOF OF LEMMA 3

The proof of the Lemma consists of the following steps:

1. Calculating $\nabla_{\boldsymbol{\theta}}^2 \mathcal{L}$, and showing that at a global minimum it takes the form $\nabla_{\boldsymbol{\theta}}^2 \mathcal{L} = \frac{1}{n} \boldsymbol{\Phi} \boldsymbol{\Phi}^\top$ (Appendix F.1).

2. Lower bounding $\lambda_{\max}(\nabla_{\boldsymbol{\theta}}^2 \mathcal{L})$ by using

$$\lambda_{\max}\left(\nabla_{\boldsymbol{\theta}}^2 \mathcal{L}\right) = \max_{\boldsymbol{v} \in \mathbb{S}^{(d+2)k}} \boldsymbol{v}^\top \left(\nabla_{\boldsymbol{\theta}}^2 \mathcal{L}\right) \boldsymbol{v} = \max_{\boldsymbol{v} \in \mathbb{S}^{(d+2)k}} \frac{1}{n} \left\|\boldsymbol{\Phi}^\top \boldsymbol{v}\right\|^2 = \max_{\boldsymbol{u} \in \mathbb{S}^{n-1}} \frac{1}{n} \left\|\boldsymbol{\Phi} \boldsymbol{u}\right\|^2,$$
(38)

and lower bounding the right hand side (Appendix F.2).

3. Simplifying the lower bound to obtain a more interpretable version which does not depend on the specific implementation of $f$ (Appendix F.3).

### F.1 HESSIAN COMPUTATION

Recall that

$$\mathcal{L}(f) = \frac{1}{2n} \sum_{j=1}^n \left(f\left(\boldsymbol{x}_j\right) - y_j\right)^2,$$
(39)

where

$$f(\boldsymbol{x}) = \sum_{i=1}^k w_i^{(2)} \sigma\left(\boldsymbol{x}^\top \boldsymbol{w}_i^{(1)} + b_i^{(1)}\right) + b^{(2)}.$$
(40)

We denote

$$\boldsymbol{W}^{(1)} = \left[\boldsymbol{w}_1^{(1)}, \cdots, \boldsymbol{w}_k^{(1)}\right] \in \mathbb{R}^{d \times k}, \quad \boldsymbol{b}^{(1)} = \left[b_1^{(1)}, \cdots, b_k^{(1)}\right]^\top \in \mathbb{R}^k,$$

$$\boldsymbol{w}^{(2)} = \left[w_1^{(2)}, \cdots, w_k^{(2)}\right]^\top \in \mathbb{R}^k, \quad b^{(2)} \in \mathbb{R}$$
(41)

and

$$\boldsymbol{\theta} = \begin{bmatrix} \text{vec}\left(\boldsymbol{W}^{(1)}\right) \\ \boldsymbol{b}^{(1)} \\ \boldsymbol{w}^{(2)} \\ b^{(2)} \end{bmatrix} \in \mathbb{R}^{(d+2)k+1}.$$
(42)

Using these notations, assuming that $\boldsymbol{\theta}^*$ is a twice differentiable global minimum of $\mathcal{L}$, we have that the gradient is

$$\nabla_{\boldsymbol{\theta}} \mathcal{L} = \frac{1}{n} \sum_{j=1}^n \left(f\left(\boldsymbol{x}_j\right) - y_j\right) \nabla_{\boldsymbol{\theta}} f\left(\boldsymbol{x}_j\right).$$
(43)

The Hessian is given by

$$\nabla_{\boldsymbol{\theta}}^2 \mathcal{L} = \frac{1}{n} \sum_{j=1}^n \nabla_{\boldsymbol{\theta}} f\left(\boldsymbol{x}_j\right) \nabla_{\boldsymbol{\theta}} f\left(\boldsymbol{x}_j\right)^\top + \frac{1}{n} \sum_{j=1}^n \left(f\left(\boldsymbol{x}_j\right) - y_j\right) \nabla_{\boldsymbol{\theta}}^2 f\left(\boldsymbol{x}_j\right)$$

$$= \frac{1}{n} \sum_{j=1}^n \nabla_{\boldsymbol{\theta}} f\left(\boldsymbol{x}_j\right) \nabla_{\boldsymbol{\theta}} f\left(\boldsymbol{x}_j\right)^\top,$$
(44)

where in the last transition we used $\forall j \in [n] : f(\boldsymbol{x}_j) = y_j$ (see Def. 2). From direct calculation we obtain

$$\nabla_{\boldsymbol{\theta}} f\left(\boldsymbol{x}\right) = \begin{pmatrix} \text{vec}\left(\frac{\partial f}{\partial \boldsymbol{W}^{(1)}}\right) \\ \nabla_{\boldsymbol{b}^{(1)}} f \\ \nabla_{\boldsymbol{w}^{(2)}} f \\ \nabla_{b^{(2)}} f \end{pmatrix} = \begin{pmatrix} \left(\boldsymbol{w}^{(2)} \odot \mathbb{I}\left(\boldsymbol{x}; \boldsymbol{\theta}\right)\right) \otimes \boldsymbol{x} \\ \boldsymbol{w}^{(2)} \odot \mathbb{I}\left(\boldsymbol{x}; \boldsymbol{\theta}\right) \\ \left(\left(\boldsymbol{W}^{(1)}\right)^\top \boldsymbol{x} + \boldsymbol{b}^{(1)}\right) \odot \mathbb{I}\left(\boldsymbol{x}; \boldsymbol{\theta}\right) \\ 1 \end{pmatrix},$$
(45)

where $\odot$ denotes the Hadamard product, $\otimes$ represents the Kronecker product and $\mathbb{I} : \mathbb{R}^d \times \mathbb{R}^{(d+2)k+1} \to \{0,1\}^k$ is the activation pattern of all neurons for input $\boldsymbol{x}$, namely $[\mathbb{I}(\boldsymbol{x}; \boldsymbol{\theta})]_i = 1$ if $\boldsymbol{x}^\top \boldsymbol{w}_i^{(1)} + b_i^{(1)} > 0$ and $[\mathbb{I}(\boldsymbol{x}; \boldsymbol{\theta})]_i = 0$ otherwise. Let us denote the tangent features matrix by

$$\boldsymbol{\Phi} = [\nabla_{\boldsymbol{\theta}} f\left(\boldsymbol{x}_1\right) \quad \nabla_{\boldsymbol{\theta}} f\left(\boldsymbol{x}_2\right) \quad \cdots \quad \nabla_{\boldsymbol{\theta}} f\left(\boldsymbol{x}_n\right)] \in \mathbb{R}^{(dk+2k+1) \times n}.$$
(46)

Then the Hessian can be expressed as $\nabla_{\boldsymbol{\theta}}^2 \mathcal{L} = \boldsymbol{\Phi} \boldsymbol{\Phi}^\top / n$, and its maximal eigenvalue can be written as

$$\lambda_{\max}(\nabla_{\boldsymbol{\theta}}^2 \mathcal{L}) = \max_{\boldsymbol{v} \in \mathbb{S}^{(d+2)k}} \boldsymbol{v}^\top \nabla_{\boldsymbol{\theta}}^2 \mathcal{L} \boldsymbol{v} = \max_{\boldsymbol{v} \in \mathbb{S}^{(d+2)k}} \frac{1}{n} \left\|\boldsymbol{\Phi}^\top \boldsymbol{v}\right\|^2 = \max_{\boldsymbol{u} \in \mathbb{S}^{n-1}} \frac{1}{n} \left\|\boldsymbol{\Phi} \boldsymbol{u}\right\|^2.$$
(47)

### F.2 LOWER BOUNDING THE TOP EIGENVALUE

Continuing from the previous section's calculation, if we take $\boldsymbol{u} = \frac{1}{\sqrt{n}}\mathbf{1}$, we obtain

$$\max_{\boldsymbol{u} \in \mathbb{S}^{n-1}} \frac{1}{n} \|\boldsymbol{\Phi}\boldsymbol{u}\|^2$$

$$\geq \frac{1}{n^2} \|\boldsymbol{\Phi}\mathbf{1}\|^2$$

$$= 1 + \frac{1}{n^2} \sum_{i=1}^{k} \left[ \sum_{l=1}^{d} \left( \sum_{j=1}^{n} w_i^{(2)} \boldsymbol{x}_{j,l} \mathbb{I}_{j,i} \right)^2 + \left( \sum_{j=1}^{n} w_i^{(2)} \mathbb{I}_{j,i} \right)^2 + \left( \sum_{j=1}^{n} \sigma\left( \boldsymbol{x}_j^\top \boldsymbol{w}_i^{(1)} + b_i^{(1)} \right) \right)^2 \right]$$

$$= 1 + \frac{1}{n^2} \sum_{i=1}^{k} \left[ \left( w_i^{(2)} \right)^2 \left( \sum_{l=1}^{d} \left( \sum_{j=1}^{n} \boldsymbol{x}_{j,l} \mathbb{I}_{j,i} \right)^2 + \left( \sum_{j=1}^{n} \mathbb{I}_{j,i} \right)^2 \right) + \left( \sum_{j=1}^{n} \sigma\left( \boldsymbol{x}_j^\top \boldsymbol{w}_i^{(1)} + b_i^{(1)} \right) \right)^2 \right]$$

$$\overset{(*)}{\geq} 1 + \frac{2}{n^2} \sum_{i=1}^{k} \left| w_i^{(2)} \right| \sqrt{\sum_{l=1}^{d} \left( \sum_{j=1}^{n} \boldsymbol{x}_{j,l} \mathbb{I}_{j,i} \right)^2 + \left( \sum_{j=1}^{n} \mathbb{I}_{j,i} \right)^2} \left| \sum_{j=1}^{n} \sigma\left( \boldsymbol{x}_j^\top \boldsymbol{w}_i^{(1)} + b_i^{(1)} \right) \right|, \qquad (48)$$

where in $(*)$ we used $\alpha^2 + \beta^2 \geq 2|\alpha\beta|$. Let $C_i \subseteq \{\boldsymbol{x}_j\}$ be the set of training points for which the $i$th neuron is active, and denote $n_i = |C_i|$, that is

$$n_i = \sum_{j=1}^{n} \mathbb{I}_{j,i}. \qquad (49)$$

Then,

$$\lambda_{\max}\left(\nabla^2_{\boldsymbol{\theta}}\mathcal{L}\right) \geq 1 + \frac{2}{n^2} \sum_{i=1}^{k} \left| w_i^{(2)} \right| \sqrt{\left\| \sum_{\boldsymbol{x} \in C_i} \boldsymbol{x} \right\|^2 + n_i^2} \left| \sum_{\boldsymbol{x} \in C_i} \left( \boldsymbol{x}_j^\top \boldsymbol{w}_i^{(1)} + b_i^{(1)} \right) \right|$$

$$= 1 + 2 \sum_{i=1}^{k} \left| w_i^{(2)} \right| \left( \frac{n_i}{n} \right)^2 \sqrt{\left\| \frac{1}{n_i} \sum_{\boldsymbol{x} \in C_i} \boldsymbol{x} \right\|^2 + 1} \left| \frac{1}{n_i} \sum_{\boldsymbol{x} \in C_i} \left( \boldsymbol{x}^\top \boldsymbol{w}_i^{(1)} + b_i^{(1)} \right) \right|$$

$$= 1 + 2 \sum_{i=1}^{k} \left| w_i^{(2)} \right| \left( \mathbb{P}\left( \boldsymbol{X} \in C_i \right) \right)^2 \sqrt{\left\| \mathbb{E}\left[ \boldsymbol{X} | \boldsymbol{X} \in C_i \right] \right\|^2 + 1} \, \mathbb{E}\left[ \boldsymbol{X}^\top \boldsymbol{w}_i^{(1)} + b_i^{(1)} | \boldsymbol{X} \in C_i \right],$$

$$(50)$$

where $\boldsymbol{X}$ is a random sample from the dataset under uniform distribution. Next, we define

$$\bar{\boldsymbol{w}}_i^{(1)} \triangleq \frac{\boldsymbol{w}_i^{(1)}}{\left\| \boldsymbol{w}_i^{(1)} \right\|}, \qquad \bar{b}_i^{(1)} \triangleq \frac{-b_i^{(1)}}{\left\| \boldsymbol{w}_i^{(1)} \right\|}. \qquad (51)$$

Using these notations we obtain

$$\lambda_{\max}\left(\nabla^2_{\boldsymbol{\theta}}\mathcal{L}\right) \geq 1 + 2 \sum_{i=1}^{k} \left| w_i^{(2)} \right| \left\| \boldsymbol{w}_i^{(1)} \right\| \left( \mathbb{P}\left( \boldsymbol{X} \in C_i \right) \right)^2 \sqrt{\left\| \mathbb{E}\left[ \boldsymbol{X} | \boldsymbol{X} \in C_i \right] \right\|^2 + 1}$$

$$\times \mathbb{E}\left[ \boldsymbol{X}^\top \bar{\boldsymbol{w}}_i^{(1)} - \bar{b}_i^{(1)} | \boldsymbol{X} \in C_i \right]$$

$$\overset{(*)}{=} 1 + 2 \sum_{i=1}^{k} \left| w_i^{(2)} \right| \left\| \boldsymbol{w}_i^{(1)} \right\| \tilde{g}\left( \bar{\boldsymbol{w}}_i^{(1)}, \bar{b}_i^{(1)} \right)$$

$$\overset{(**)}{\geq} 1 + 2 \sum_{i=1}^{k} \left| w_i^{(2)} \right| \left\| \boldsymbol{w}_i^{(1)} \right\| g\left( \bar{\boldsymbol{w}}_i^{(1)}, \bar{b}_i^{(1)} \right), \qquad (52)$$

where in $(*)$ and $(**)$ we defined, respectively,

$$
\tilde{g}\left(\bar{\boldsymbol{w}}, \bar{b}\right) = \left(\mathbb{P}\left(\boldsymbol{X}^\top \bar{\boldsymbol{w}} > \bar{b}\right)\right)^2 \mathbb{E}\left[\boldsymbol{X}^\top \bar{\boldsymbol{w}} - \bar{b} \middle| \boldsymbol{X}^\top \bar{\boldsymbol{w}} > \bar{b}\right] \sqrt{\left\|\mathbb{E}\left[\boldsymbol{X}\middle|\boldsymbol{X}^\top \bar{\boldsymbol{w}} > \bar{b}\right]\right\|^2 + 1},
$$
$$
g\left(\bar{\boldsymbol{w}}, \bar{b}\right) = \min\left(\tilde{g}\left(\bar{\boldsymbol{w}}, \bar{b}\right), \tilde{g}\left(-\bar{\boldsymbol{w}}, -\bar{b}\right)\right). \tag{53}
$$

From our derivation so far, we obtain that

$$
\lambda_{\max}\left(\nabla_{\boldsymbol{\theta}}^2 \mathcal{L}\right) \geq 1 + 2 \sum_{i=1}^{k} \left|w_i^{(2)}\right| \left\|\boldsymbol{w}_i^{(1)}\right\| g\left(\bar{\boldsymbol{w}}_i^{(1)}, \bar{b}_i^{(1)}\right). \tag{54}
$$

We denote $a_i = w_i^{(2)} \|\boldsymbol{w}_i^{(1)}\|$ and the representation dependent stability norm as

$$
S_{\boldsymbol{\theta}} \triangleq \sum_{i=1}^{k} |a_i| \, g\left(\bar{\boldsymbol{w}}_i^{(1)}, \bar{b}_i^{(1)}\right). \tag{55}
$$

### F.3   IMPLEMENTATION FREE LOWER BOUND

In this section, our goal is to give a simpler lower bound of the multivariate stability norm $S_{\boldsymbol{\theta}}$, that does not depend on the specific representation of $f$. Let $\alpha$ be the signed measure over $\mathbb{S}^{d-1} \times \mathbb{R}$ given by $\alpha = \sum_{i=1}^{k} a_i \delta_{\left(\bar{\boldsymbol{w}}_i^{(1)}, \bar{b}_i^{(1)}\right)}$, and whose total variation measure $|\alpha|$ is given by $|\alpha| = \sum_{i=1}^{k} |a_i| \delta_{\left(\bar{\boldsymbol{w}}_i^{(1)}, \bar{b}_i^{(1)}\right)}$. Recall that

$$
f(\boldsymbol{x}) = \sum_{i=1}^{k} \left\|\boldsymbol{w}_i^{(1)}\right\| w_i^{(2)} \sigma\left(\boldsymbol{x}^\top \bar{\boldsymbol{w}}_i^{(1)} - \bar{b}_i^{(1)}\right) + b^{(2)} = \sum_{i=1}^{k} a_i \sigma\left(\boldsymbol{x}^\top \bar{\boldsymbol{w}}_i^{(1)} - \bar{b}_i^{(1)}\right) + b^{(2)}, \tag{56}
$$

and thus

$$
\begin{aligned}
\Delta f(\boldsymbol{x}) &= \sum_{l=1}^{d} \frac{\partial^2 f(\boldsymbol{x})}{\partial x_l^2} \\
&= \sum_{i=1}^{k} a_i \delta\left(\boldsymbol{x}^\top \bar{\boldsymbol{w}}_i^{(1)} - \bar{b}_i^{(1)}\right) \\
&= \int_{\mathbb{S}^{d-1} \times \mathbb{R}} \alpha\left(\bar{\boldsymbol{w}}, \bar{b}\right) \delta\left(\bar{\boldsymbol{w}}^\top \boldsymbol{x} - \bar{b}\right) \mathrm{d}s(\bar{\boldsymbol{w}})\mathrm{d}\bar{b} \\
&= \int_{\mathbb{S}^{d-1}} \alpha\left(\bar{\boldsymbol{w}}, \bar{\boldsymbol{w}}^\top \boldsymbol{x}\right) \mathrm{d}s(\bar{\boldsymbol{w}}) \\
&= \mathcal{R}^* \alpha. 
\end{aligned} \tag{57}
$$

Namely, $\Delta f$ is a weighted sum of Diracs supported on hyperplanes. From the last equation we obtain $\alpha = (\mathcal{R}^*)^{-1} \Delta f$. Combining these results we get that

$$
S_{\boldsymbol{\theta}} = \int_{\mathbb{S}^{d-1} \times \mathbb{R}} g \mathrm{d}|\alpha| = \langle |\alpha|, g\rangle \geq \int_{\mathbb{S}^{d-1} \times \mathbb{R}} \left|\left[(\mathcal{R}^*)^{-1} \Delta f\right](v, b)\right| g(v, b) \mathrm{d}s(v)\mathrm{d}b = \|f\|_{\mathcal{R}, g}, \tag{58}
$$

where the inequality in the third step is due to $g$ being non-negative and the scenarios in which multiple deltas become active at the same location, namely $\exists i \neq j : \bar{\boldsymbol{w}}_i^{(1)} = \bar{\boldsymbol{w}}_j^{(1)}$ and $\bar{b}_i^{(1)} = \bar{b}_j^{(1)}$. Note that if the deltas do not align, then $S_{\boldsymbol{\theta}} = \|f\|_{\mathcal{R}, g}$. Overall we have that

$$
\lambda_{\max}\left(\nabla_{\boldsymbol{\theta}}^2 \mathcal{L}\right) \geq 1 + 2 S_{\boldsymbol{\theta}} \geq 1 + 2 \|f\|_{\mathcal{R}, g}. \tag{59}
$$

## G   DERIVATION OF THE STABILITY NORM IN PRIMAL SPACE

We defined the multivariate stability norm as $\|f\|_{\mathcal{R}, g} = \langle |(\mathcal{R}^*)^{-1} \Delta f|, g\rangle_{\mathbb{S}^{d-1} \times \mathbb{R}}$, where $\langle \cdot, \cdot \rangle_{\mathbb{S}^{d-1} \times \mathbb{R}}$ denotes the integral inner-product on $\mathbb{S}^{d-1} \times \mathbb{R}$. Supposing that the inverse Radon transform of $g$

exists, then by purely formal reasoning we ought to have

$$
\begin{aligned}
\|f\|_{\mathcal{R},g} &= \left\langle \left| (\mathcal{R}^*)^{-1} \Delta f \right|, g \right\rangle_{\mathbb{S}^{d-1} \times \mathbb{R}} \\
&= \left\langle \left| (\mathcal{R}^*)^{-1} \Delta f \right|, \mathcal{R}\mathcal{R}^{-1} g \right\rangle_{\mathbb{S}^{d-1} \times \mathbb{R}} \\
&= \left\langle \mathcal{R}^* \left| (\mathcal{R}^*)^{-1} \Delta f \right|, \mathcal{R}^{-1} g \right\rangle_{\mathbb{R}^d} .
\end{aligned}
\tag{60}
$$

Therefore, making the (formal) definitions $|\Delta f|_{\mathcal{R}} \triangleq \mathcal{R}^* |(\mathcal{R}^*)^{-1} \Delta f|$ and $\rho \triangleq \mathcal{R}^{-1} g$, we may also interpret the stability norm $\|f\|_{\mathcal{R},g}$ as the quantity

$$
\int_{\mathbb{R}^d} |\Delta f|_{\mathcal{R}}(\boldsymbol{x}) \rho(\boldsymbol{x}) \mathrm{d}\boldsymbol{x}.
\tag{61}
$$

In the event that $f$ and $g$ are smooth, and $g$ is in the range of the classical Radon transform, then the above expression is equal to $\|f\|_{\mathcal{R},g}$. However, this is not generally the case in our setting, and below we show how to give a more precise interpretation of this integral formula using distributional theory. In particular, we show that when $f$ is a finite-width ReLU network $|\Delta f|_{\mathcal{R}}$ is equal to the total variation measure of $\Delta f$ (*i.e.*, the measure-theoretic analog of the absolute value of a function). Additionally, in the event that $g$ does not have a classically defined Radon inverse, we show how the integral in (61) can be interpreted using a smoothing approach.

Let $f$ be a finite width single hidden-layer ReLU network, *i.e.*, $f \in \mathcal{F}_k$ for some finite $k$. Recall that $(\mathcal{R}^*)^{-1} \Delta f$ is a finite weighted sum of Diracs in $\mathbb{S}^{d-1} \times \mathbb{R}$. Let $|(\mathcal{R}^*)^{-1} \Delta f|$ be the associated total variation measure, and define $|\Delta f|_{\mathcal{R}} = \mathcal{R}^* |(\mathcal{R}^*)^{-1} \Delta f|$, where $\mathcal{R}^*$ is the distributional dual Radon transform. Here $|\Delta f|_{\mathcal{R}}$ is a tempered distribution given by a (positive) weighted sum of Diracs supported on hyperplanes. For example, if $f$ is a single ReLU unit of the form $f(\boldsymbol{x}) = a\,\sigma(\boldsymbol{x}^\top \boldsymbol{v} - b)$ with $\|\boldsymbol{v}\|_2 = 1$, then $|\Delta f|_{\mathcal{R}}$ is the distribution $|a|\delta(\boldsymbol{x}^\top \boldsymbol{v} - b)$, that is, for any test function $\phi$ we have

$$
\langle |\Delta f|_{\mathcal{R}}, \phi \rangle_{\mathbb{R}^d} = |a| \int_{\boldsymbol{x}^\top \boldsymbol{v} = b} \phi(\boldsymbol{x}) \mathrm{d}s(\boldsymbol{x}) = |a| \mathcal{R}\phi(\boldsymbol{v}, b).
\tag{62}
$$

On the other hand, treating $\Delta f$ as a measure defined over a compact subset of $\mathbb{R}^d$, its total variation measure $|\Delta f|$ is also equal to $|a|\delta(\boldsymbol{x}^\top \boldsymbol{v} - b)$. The following result shows that, more generally, when $f$ is any finite width ReLU net then $|\Delta f|_{\mathcal{R}}$ and $|\Delta f|$ are equal as measures.

**Proposition 4.** *Let $f \in \mathcal{F}_k$, then $|\Delta f|_{\mathcal{R}} = |\Delta f|$ as measures defined over any compact set of $\mathbb{R}^d$.*

*Proof.* Since $f \in \mathcal{F}_k$, there exists a representation of $f$ as $f(\boldsymbol{x}) = \sum_{i=1}^{k'} a_i \sigma(\boldsymbol{v}_i^\top \boldsymbol{x} - b_i) + \boldsymbol{x}^\top \boldsymbol{q} + c$ where $\|\boldsymbol{v}_i\| = 1$ for all $i$, $a_i \neq 0$ for all $i$, and $(\boldsymbol{v}_i, b_i) \neq \pm(\boldsymbol{v}_j, b_j)$ for all $i \neq j$ (*i.e.*, the knots of all ReLU units are distinct[8]), and where $k' \leq k$. Therefore, each ReLU unit in this representation of $f$ maps to a distinct Dirac $\delta_{(\boldsymbol{v}_i, b_i)}$ in Radon space after applying the operator $(\mathcal{R}^*)^{-1}\Delta$, and so

$$
|(\mathcal{R}^*)^{-1} \Delta f| = \sum_{i=1}^{k'} |a_i| \delta_{(\boldsymbol{v}_i, b_i)}.
\tag{63}
$$

Let $X$ be any compact subset of $\mathbb{R}^d$, and let $\phi$ be any continuous test function defined over $X$. Then,

$$
\langle |\Delta f|_{\mathcal{R}}, \phi \rangle = \langle |(\mathcal{R}^*)^{-1} \Delta f|, \mathcal{R}\phi \rangle = \sum_{i=1}^{k'} |a_i| \mathcal{R}\phi(\boldsymbol{v}_i, b_i).
\tag{64}
$$

Now, we show the same equality holds with $|\Delta f|$ in place of $|\Delta f|_{\mathcal{R}}$. Let $I^+$ denote the set of indices $i$ such that $a_i > 0$ and $I^-$ denote the set of indices $i$ such that $a_i < 0$. Define the measures $\mu_+ = \sum_{i \in I^+} a_i \delta(\boldsymbol{v}_i^\top \cdot - b_i)$ and $\mu_- = -\sum_{i \in I^-} a_i \delta(\boldsymbol{v}_i^\top \cdot - b_i)$. Observe that $\mu_+$ and $\mu_-$ are both positive measures whose supports only possibly intersect on a set of measure zero, and $\Delta f = \mu_+ - \mu_-$. This implies the total variation measure of $\Delta f$ is given by $|\Delta f| = \mu_+ + \mu_- = \sum_{i=1}^{k'} |a_i| \delta(\boldsymbol{v}_i^\top \cdot - b_i)$. Hence,

$$
\langle |\Delta f|, \phi \rangle = \sum_{i=1}^{k'} |a_i| \mathcal{R}\phi(\boldsymbol{v}_i, b_i),
\tag{65}
$$

as claimed. $\qquad\qquad\square$

---

[8]If $(\boldsymbol{v}_i, b_i) = (\boldsymbol{v}_j, b_j)$ then one of the neurons is redundant. If $(\boldsymbol{v}_i, b_i) = -(\boldsymbol{v}_j, b_j)$ then these units can be combined into an affine function, which we "absorb" into the term $\boldsymbol{x}^\top \boldsymbol{q} + c$.

Note, however, that when $f$ corresponds to finite-width ReLU network, $|\Delta f|$ does not have finite total variation considered as a measure defined over all $\mathbb{R}^d$. Due to this technicality, when $\rho$ is not compactly supported, we need to understand the integral in (61) as being with respect to the distribution $|\Delta f|_\mathcal{R}$ in place of the measure $|\Delta f|$.

Now we show that when $\rho = \mathcal{R}^{-1}g$ does not exist in a classical sense, the integral in (61) can still be interpreted using a smoothing approach.

**Proposition 5.** *Let $f \in \mathcal{F}_k$ and suppose $g$ is an even, piecewise continuous $L^1$ function on $\mathbb{S}^{d-1} \times \mathbb{R}$ and let $\rho = \mathcal{R}^{-1}g$ be its distibutional Radon inverse. Further, assume the support of $|(\mathcal{R}^*)^{-1}\Delta f|$ does not intersect the set of points where $g$ is discontinuous. Then $\|f\|_{\mathcal{R},g}$ is finite and*

$$\|f\|_{\mathcal{R},g} = \int_{\mathbb{R}^d} |\Delta f(\boldsymbol{x})|\rho(\boldsymbol{x})\mathrm{d}\boldsymbol{x}, \tag{66}$$

*where the integral above is understood as the finite limit*

$$\lim_{\epsilon \to 0} \langle |\Delta f|_\mathcal{R}, \rho_\epsilon \rangle_{\mathbb{R}^d}, \tag{67}$$

*where $\rho_\epsilon$ is a smooth approximation of $\rho$ defined independently of $f$ whose classical Radon transform $g_\epsilon = \mathcal{R}\rho_\epsilon$ exists for all $\epsilon > 0$, and $g_\epsilon \to g$ uniformly as $\epsilon \to 0$ on any closed subset of $\mathbb{R}^d$ over which $g$ is continuous.*

*Proof.* For any $\epsilon > 0$ let $\phi_\epsilon \in \mathcal{S}(\mathbb{S}^{d-1} \times \mathbb{R})$ be a compactly supported even function acting as a smooth approximation of the identity, *i.e.*, for any continuous, even function $h$ vanishing at infinity we have $\phi_\epsilon * h \to h$ uniformly as $\epsilon \to 0$, where $*$ denotes convolution of functions on $\mathbb{S}^{d-1} \times \mathbb{R}$. Define $g_\epsilon = (\phi_\epsilon * g) \cdot \chi_\epsilon$, where $\chi_\epsilon(\boldsymbol{v}, b)$ is a smooth cutoff function that is equal to one if $|b| \leq 1/\epsilon$ and rapidly decays to zero for $|b| \geq 1/\epsilon$. Observe that $g_\epsilon$ is an even Schwartz function by construction. Furthermore, since $g$ is piecewise continuous and $L^1$ (and in particular, it vanishes at infinity), this implies $g_\epsilon \to g$ uniformly over any closed set that does not intersect the set of points where $g$ is discontinuous. Therefore, if we let $U \subset \mathbb{S}^{d-1} \times \mathbb{R}$ be any closed set containing the support of $|(\mathcal{R}^*)^{-1}\Delta f|$ that does not intersect the set of points where $g$ is discontinuous (which is guaranteed to exist since the support of $|(\mathcal{R}^*)^{-1}\Delta f|$ is a finite set), then we have $g_\epsilon \to g$ uniformly over $U$. Therefore

$$\|f\|_{\mathcal{R},g} = \langle |(\mathcal{R}^*)^{-1}\Delta f|, g \rangle_{\mathbb{S}^{d-1}\times\mathbb{R}} = \lim_{\epsilon\to 0}\langle |(\mathcal{R}^*)^{-1}\Delta f|, g_\epsilon \rangle_{\mathbb{S}^{d-1}\times\mathbb{R}}, \tag{68}$$

where the limit is guaranteed to exist since the finite measure $|(\mathcal{R}^*)^{-1}\Delta f|$ is a continuous linear functional over $C_0(U)$, the space of continuous functions over $U$ vanishing at infinity. Finally, since $g_\epsilon$ is Schwartz, (Solmon, 1987, Thm. 7.7) guarantees $\rho_\epsilon = \mathcal{R}^{-1}g_\epsilon$ exists as a $C^\infty$-smooth function on $\mathbb{R}^d$ that is also integrable along hyperplanes and for which the classical Radon inversion formula holds: $\mathcal{R}\rho_\epsilon = g_\epsilon$. Therefore, we have

$$\begin{aligned}\|f\|_{\mathcal{R},g} &= \lim_{\epsilon\to 0}\langle |(\mathcal{R}^*)^{-1}\Delta f|, \mathcal{R}\rho_\epsilon \rangle_{\mathbb{S}^{d-1}\times\mathbb{R}} \\ &= \lim_{\epsilon\to 0}\langle \mathcal{R}^*|(\mathcal{R}^*)^{-1}\Delta f|, \rho_\epsilon \rangle_{\mathbb{R}^d} \\ &= \lim_{\epsilon\to 0}\langle |\Delta f|_\mathcal{R}, \rho_\epsilon \rangle_{\mathbb{R}^d}, \end{aligned} \tag{69}$$

as claimed. $\qquad\square$

The assumption made above that the support of $|(\mathcal{R}^*)^{-1}\Delta f|$ does not intersect the set of points where $g$ is not overly restrictive. For example, this assumption holds when $f$ corresponds to a differentiable minimizer of the squared loss defined in terms of a finite set of training points and $g$ is the data dependent weighting function defined in (12). In this case, the discontinuity set of $g(\boldsymbol{v}, b)$ corresponds to the set of hyperplanes $\{\boldsymbol{x} : \mathbb{R}^d : \boldsymbol{x}^\top \boldsymbol{v} = b\}$ that intersect one or more of the training points. And $f$ corresponds to a differentiable minimizer if and only if the hyperplanes defined by the knots of the ReLU units making up $f$ (*i.e.*, the support of $|(\mathcal{R}^*)^{-1}f|$) do not intersect any training points.

## H   EXAMPLES OF $g$ AND $\rho$

### H.1   TWO DATAPOINTS

For this example, it is easy to calculate that

$$g(\boldsymbol{v}, b) = \alpha\,\sigma\big(|v_1| - |b|\big), \tag{70}$$

where $\alpha$ is a positive constant. We compute $\rho = \mathcal{R}^{-1}g$ by first determining its Laplacian, $\Delta\rho$, then inverting and Laplacian to recover $\rho$. First, the intertwining property of the Laplacian and the Radon transform gives

$$\mathcal{R}\Delta\rho = \frac{\partial}{\partial b^2}\mathcal{R}\rho = \frac{\partial}{\partial b^2}g. \tag{71}$$

Therefore, by the Fourier slice theorem (Helgason, 1999), for all $(\boldsymbol{v}, s) \in \mathbb{S}^{d-1} \times \mathbb{R}$ we have

$$\mathfrak{F}\{\Delta\rho\}(s\boldsymbol{v}) = \mathfrak{F}_b\left\{\frac{\partial}{\partial b^2}g\right\}(\boldsymbol{v}, s), \tag{72}$$

where $\mathfrak{F}\{\cdot\}$ is the 2-D Fourier transform, and $\mathfrak{F}_b\{\cdot\}$ is the Fourier transform in the $b$ variable. For fixed $\boldsymbol{v}$, the function $b \to g(\boldsymbol{v}, b)$ is continuous and piecewise linear with knots at $0$ and $\pm|v_1|$, and it is easy to see that

$$\frac{\partial}{\partial b^2}g(\boldsymbol{v}, b) = \alpha\big(\delta(b - |v_1|) + \delta(b + |v_1|) - 2\delta(b)\big), \tag{73}$$

which implies

$$\mathfrak{F}\{\Delta\rho\}(s\boldsymbol{v}) = \mathfrak{F}_b\left\{\frac{\partial}{\partial b^2}g\right\}(\boldsymbol{v}, s) = \alpha\left(e^{-j2\pi|v_1|s} + e^{j2\pi|v_1|s} - 2\right). \tag{74}$$

If we restrict the unit-norm vector $\boldsymbol{v} = (v_1, v_2)$ to be such that $v_1 \geq 0$ and define $\boldsymbol{\xi} = s\boldsymbol{v}$ then we see that $\boldsymbol{x}_1^\top\boldsymbol{\xi} = s|v_1|$ and $\boldsymbol{x}_2^\top\boldsymbol{\xi} = -s|v_1|$. Therefore, we have

$$\mathfrak{F}\{\Delta\rho\}(\boldsymbol{\xi}) = \alpha\left(e^{-j2\pi\boldsymbol{x}_1^\top\boldsymbol{\xi}} + e^{-j2\pi\boldsymbol{x}_2^\top\boldsymbol{\xi}} - 2\right), \tag{75}$$

and inverting the Fourier transform gives

$$\Delta\rho(\boldsymbol{x}) = \alpha\left(\delta(\boldsymbol{x} - \boldsymbol{x}_1) + \delta(\boldsymbol{x} - \boldsymbol{x}_2) - 2\delta(\boldsymbol{x})\right). \tag{76}$$

Finally, since $\varphi(\boldsymbol{x}) = \frac{1}{2\pi}\log(\|\boldsymbol{x}\|)$ is the fundamental solution of Poisson's equation in 2D (*i.e.*, $\Delta\varphi = \delta$, where $\delta$ is a Dirac centered at origin), we have

$$\rho(\boldsymbol{x}) = \frac{\alpha}{2\pi}\big(\log(\|\boldsymbol{x} - \boldsymbol{x}_1\|) + \log(\|\boldsymbol{x} - \boldsymbol{x}_2\|) - 2\log(\|\boldsymbol{x}\|)\big). \tag{77}$$

While each term in the sum above not absolutely integrable along lines, their sum is absolutely integrable along lines. This is because the function $t \to \log(|t|)$ is absolutely integrable over any neighborhood of the origin, and by a multipole expansion we may show that $\rho(\boldsymbol{x}) = O(\|\boldsymbol{x}\|^{-2})$ as $\boldsymbol{x} \to \infty$, which is also absolutely integrable along lines.

### H.2 ISOTROPIC DATA DISTRIBUTION

For an isotropic data distribution, *i.e.*, $\mathbb{P}\left(\boldsymbol{x}^\top\boldsymbol{v} > b\right) = M(b)$ for any $\boldsymbol{v}$ that satisfies $\|\boldsymbol{v}\| = 1$, we will have that

$$\tilde{g}(\boldsymbol{v}, b) = M(b)\int_b^\infty M(z)\mathrm{d}z\sqrt{\left(b + \frac{1}{M(b)}\int_b^\infty M(z)\mathrm{d}z\right)^2 + 1}. \tag{78}$$

Then, from symmetry and assuming that $\tilde{g}$ is decreasing in $b$ we obtain

$$g(\boldsymbol{v}, b) = M(|b|)\int_{|b|}^\infty M(z)\mathrm{d}z\sqrt{\left(|b| + \frac{1}{M(|b|)}\int_{|b|}^\infty M(z)\mathrm{d}z\right)^2 + 1}. \tag{79}$$

Note that $g$ only depends on $|b|$ and is decreasing in $|b|$. Considering the parameter space representation for the stability norm

$$S_{\boldsymbol{\theta}} = \sum_{i=1}^k |a_i| g(\boldsymbol{v}_i, b_i), \tag{80}$$

we can see that solutions with larger $|b_i|$, *i.e.*, solutions which are more flat in function space, will have smaller stability norm.

We now characterize $\rho = \mathcal{R}^{-1}g$. For simplicity, we focus on the two-dimensional setting ($d = 2$). Since $g(\boldsymbol{v}, b)$ does not depend on $\boldsymbol{v}$, we drop this dependence and simply write $g(b)$. Note that this implies $\rho$ is a radial function. Let $\tilde{\rho}(r)$ denote the radial profile of $\rho$, *i.e.*, $\rho(\boldsymbol{x}) = \tilde{\rho}(\|\boldsymbol{x}\|)$. We additionally make the following assumptions: $g(b)$ is twice continuously differentiable away from the origin, and both $g$ and its weak derivative $g'$ are bounded and absolutely integrable. In this case, $\tilde{\rho}$ has the integral formula [9]

$$\tilde{\rho}(r) = -\frac{1}{\pi} \int_r^\infty \frac{g'(b)}{\sqrt{b^2 - r^2}} \mathrm{d}b. \tag{81}$$

The assumptions on $g$ above are sufficient to show the integrand in (81) is absolutely integrable over $[r, \infty)$ for $r > 0$. Since $g(|b|)$ is assumed to be decreasing in $|b|$, we have $-g'(b) \geq 0$ for all $b > 0$, which shows $\tilde{\rho}(r) \geq 0$ for all $r > 0$. However, if $g$ is not smooth at the origin, then $g'(b) = O(1)$ as $b \to 0^+$, and elementary analysis shows $\tilde{\rho}(r) = O(\log(r))$ as $r \to 0^+$ and $\tilde{\rho}(r) = O(1/r)$ as $r \to +\infty$.

Finally, if we additionally assume $g'(b)$ is non-increasing for $b > 0$, then $\rho(r)$ is strictly decreasing for $r > 0$. To see this, fix any $r' > r$, and define $\delta = r' - r$. Using the change of variables $b \mapsto b - \delta$, we may show

$$\tilde{\rho}(r') = -\frac{1}{\pi} \int_r^\infty \frac{g'(b + \delta)}{\sqrt{(b + \delta)^2 - r^2}} \mathrm{d}b. \tag{82}$$

Since $g'$ is assumed to be non-increasing, we have $g'(b + \delta) \leq g'(b)$ and it is elementary to show $((b + \delta)^2 - r^2)^{-1/2} < (b^2 - r^2)^{-1/2}$ for all $b > r$, which shows the integrand in (82) is pointwise strictly bounded above by the integrand in (81) for all $b > r$, hence $\tilde{\rho}(r') < \tilde{\rho}(r)$.

In the case of 2D Gaussian distributed data $\boldsymbol{X} \sim \mathcal{N}(\boldsymbol{0}, \boldsymbol{I})$, then $M(b)$ is the complementary of the CDF of a normal random variable: $M(b) = \frac{1}{\sqrt{2\pi}} \int_b^\infty e^{-b^2/2} \mathrm{d}b$. It is easy to verify that the resulting $g$ function satisfies the above assumptions ($g(b)$ is decreasing, twice continuously differentiable away from the origin, both $g$ and its weak derivative $g'$ are bounded and absolutely integrable, and $g'$ is non-increasing). Therefore, the resulting $\rho$ has the all the properties outlined above.

## I  DEPTH SEPARATION PROOFS

Before giving the proofs in this section we introduce some additional notation. Let $\overline{X}$ denote the closed convex hull of the training points and $X$ its open interior. Additionally, let $Y = \{(\boldsymbol{v}, b) \in \mathbb{S}^{d-1} \times \mathbb{R} : \boldsymbol{v}^\top \boldsymbol{x} > b$ for some $\boldsymbol{x} \in X\}$, and let $\overline{Y}$ denote its closure. Note that for any smooth function $\phi$ with support contained in $\overline{X}$, the Radon transform $\mathcal{R}\phi$ has support contained in $\overline{Y}$. Finally, for any distribution $h$ and open set $U$, we let $h|_U$ denote its restriction to $U$.

### I.1  PROOF OF PROPOSITION 1

First, we show that the convergence of a sequence of functions $f_k$ to $f$ in $L^1$-norm over $X$ implies that the sequence of distributions $\Delta f_k|_X$ converges weakly to the distribution $\Delta f|_X$. For all test functions $\phi \in \mathcal{S}(X)$ we have

$$|\langle \Delta f_k - \Delta f, \phi \rangle| = |\langle f_k - f, \Delta\phi \rangle| \leq \|f_k - f\|_{L^1(X)} \|\Delta\phi\|_{L^\infty(X)}, \tag{83}$$

where we used Holder's inequality to achieve the final bound. Therefore, we have $\lim_{k\to\infty} \langle \Delta f_k, \phi \rangle \to \langle \Delta f, \phi \rangle$, which proves the weak convergence.

Next, we show $(\mathcal{R}^*)^{-1}\Delta f_k|_Y$ converges weakly to $(\mathcal{R}^*)^{-1}\Delta f|_Y$. For all test functions $\varphi \in \mathcal{S}_H(Y)$ we have

$$\left\langle (\mathcal{R}^*)^{-1}\Delta f_k - (\mathcal{R}^*)^{-1}\Delta f, \varphi \right\rangle = \left\langle \Delta f_k - \Delta f, \mathcal{R}^{-1}\varphi \right\rangle. \tag{84}$$

Since $\phi$ vanishes outside $Y$, by the support theorem (Helgason, 1999, Corollary 2.8) we are ensured that $\mathcal{R}^{-1}\varphi$ has support contained in $\overline{X}$, hence $\mathcal{R}^{-1}\varphi \in \mathcal{S}(X)$. The desired result now follows immediately by weak convergence of $\Delta f_k|_X$ to $\Delta f|_X$.

Since $g$ has support contained in $\overline{Y}$, this further implies that the distribution $g \cdot (\mathcal{R}^*)^{-1}\Delta f_k$ converges weakly to the distribution $g \cdot (\mathcal{R}^*)^{-1}\Delta f$. Finally, since $\|f_k\|_{\mathcal{R},g} = \|g \cdot (\mathcal{R}^*)^{-1}\Delta f_k\|_{\mathrm{TV}}$ is bounded

---

[9] See Proposition 3.5.1 in (Epstein, 2007).

by assumption, this implies each $g \cdot (\mathcal{R}^*)^{-1} \Delta f_k$ is a measure having finite total variation, hence their weak limit $g \cdot (\mathcal{R}^*)^{-1} \Delta f$ is also a measure with finite total variation (*i.e.*, the weak limit of order-0 distributions that are bounded in TV-norm is also an order-0 distribution). Therefore, $\|f\|_{\mathcal{R},g} = \|g \cdot (\mathcal{R}^*)^{-1} \Delta f\|_{\mathrm{TV}}$ is finite, as claimed.

## I.2   Proof of Proposition 2

To show the pyramid function $p$ has infinite stability norm, we prove that $g \cdot (\mathcal{R}^*)^{-1} \Delta p$ is a distribution of order $> 0$, which implies

$$\|p\|_{\mathcal{R},g} = \sup_{\phi \in \mathcal{S}_H(\mathbb{S}^{d-1} \times \mathbb{R})} \langle g \cdot (\mathcal{R}^*)^{-1} \Delta p, \phi \rangle = +\infty. \tag{85}$$

First, observe that the Laplacian $\Delta p$ is an order-0 distribution whose support is contained in the unit $\ell^1$-ball. This implies the distribution $(\mathcal{R}^*)^{-1} \Delta p$, is supported on a compact set $K$ in Radon domain. By our assumption on the convex hull of the training points, $g(\boldsymbol{v}, b) > 0$ for all $(\boldsymbol{v}, b) \in K$. Since $g$ is piecewise continuous and $K$ is compact, this implies there exists constants $c_1, c_2 > 0$ such that $c_1 \leq g(\boldsymbol{v}, b) \leq c_2$ for all $(\boldsymbol{v}, b) \in K$. Therefore, we see that $g \cdot (\mathcal{R}^*)^{-1} \Delta p$ is an order-0 distribution if and only if $(\mathcal{R}^*)^{-1} \Delta p$ is an order-0 distribution. However, by a result in (Ongie et al., 2020), we know $(\mathcal{R}^*)^{-1} \Delta p$ has order $> 0$ (*i.e.*, in the terminology of (Ongie et al., 2020), $p$ has infinite $\mathcal{R}$-norm). Additionally, we show this by direct calculation in App. M in the case of input dimension $d = 2$. Therefore, $g \cdot (\mathcal{R}^*)^{-1} \Delta p$ must be a distribution of order $> 0$ and hence $\|p\|_{\mathcal{R},g} = +\infty$ as claimed.

## I.3   Stability of the two hidden-layer implementation of $p(\boldsymbol{x})$

Let us focus on the under-parameterized setting, in which there exists a single optimal input-output predictor $p(\boldsymbol{x})$ that globally minimizes the loss. In this case, the set of all global minima corresponds to different implementations of $p(\boldsymbol{x})$. Under this setting, we will prove that there exists a set of nonzero Lebesgue measure such that for any initialization inside this set, GD necessarily converges to $p(\boldsymbol{x})$.

To do so, we will first prove that for any minimum point $\boldsymbol{\theta}^* \in \mathbb{R}^m$ corresponding to an implementation of $p(\boldsymbol{x})$, there exists a nonzero step size $\eta$ with which $\boldsymbol{\theta}^*$ is linearly stable. Furthermore, we will show that there exists a set $\mathcal{T}_{\mathrm{loc}}^{\mathrm{s}}(\boldsymbol{\theta}^*)$ embedded in a subspace of dimension $m - m_{\mathrm{Null}}$, in which any initialization converges to $\boldsymbol{\theta}^*$. Here $m$ is the number of parameters in our two hidden-layer network, and $m_{\mathrm{Null}}$ is the number of zero eigenvalues of $\nabla^2 \mathcal{L}$ at $\boldsymbol{\theta}^*$. Next, we will show that there is a connected set of minima $\Theta^*$ around $\boldsymbol{\theta}^*$, such that the union $\bigcup_{\boldsymbol{\theta} \in \Theta^*} \mathcal{T}_{\mathrm{loc}}^{\mathrm{s}}(\boldsymbol{\theta})$ has a nonzero Lebesgue measure.

Let us start with some minimum point $\boldsymbol{\theta}^*$, which corresponds to an implementation of $p(\boldsymbol{x})$. GD's update rule is

$$\boldsymbol{\theta}_{t+1} = \boldsymbol{\theta}_t - \eta \nabla \mathcal{L}(\boldsymbol{\theta}_t). \tag{86}$$

Define the mapping

$$T(\boldsymbol{\theta}) = \boldsymbol{\theta} - \eta \nabla \mathcal{L}(\boldsymbol{\theta}). \tag{87}$$

Then (86) can be written as

$$\boldsymbol{\theta}_{t+1} = T(\boldsymbol{\theta}_t). \tag{88}$$

This equation describes the full dynamics of GD using the nonlinear mapping $T$. Note that in this representation, $\boldsymbol{\theta}^*$ is an equilibrium point of $T$, *i.e.*, $T(\boldsymbol{\theta}^*) = \boldsymbol{\theta}^*$. We would like to show that it is possible to converge to $\boldsymbol{\theta}^*$. Assume there is a finite number of training samples $n$, and none of them coincide with the knots of $p$. Then $T$ is differentiable in a small neighborhood of $\boldsymbol{\theta}^*$. The Jacobian matrix of $T$ is

$$\frac{\partial}{\partial \boldsymbol{\theta}} T = \boldsymbol{I} - \eta \nabla^2 \mathcal{L}(\boldsymbol{\theta}^*), \tag{89}$$

and its eigenvalues are

$$\lambda_i \left( \frac{\partial}{\partial \boldsymbol{\theta}} T \right) = \lambda_i \left( \boldsymbol{I} - \eta \nabla^2 \mathcal{L}(\boldsymbol{\theta}^*) \right) = 1 - \eta \lambda_i \left( \nabla^2 \mathcal{L}(\boldsymbol{\theta}^*) \right). \tag{90}$$

In this setting, the loss' Hessian at $\boldsymbol{\theta}^*$ has non-negative and bounded eigenvalues. Particularly, there exists[10] a sufficiently small step size $\eta$ that satisfies

$$0 < \left| 1 - \eta \lambda_i \left( \nabla^2 \mathcal{L}(\boldsymbol{\theta}^*) \right) \right| \leq 1, \tag{91}$$

for all $i$. Using the Center and Stable Manifold Theorem (Shub, 2013, Th. III.7), there exists a bounded set $\mathcal{T}_{\text{loc}}^{\text{s}}(\boldsymbol{\theta}^*)$ such that

$$T(\mathcal{T}_{\text{loc}}^{\text{s}}(\boldsymbol{\theta}^*)) \subseteq \mathcal{T}_{\text{loc}}^{\text{s}}(\boldsymbol{\theta}^*) \qquad \text{and} \qquad \forall \boldsymbol{\theta} \in \mathcal{T}_{\text{loc}}^{\text{s}}(\boldsymbol{\theta}^*): \ \|T(\boldsymbol{\theta}) - \boldsymbol{\theta}^*\| \leq \alpha \|\boldsymbol{\theta} - \boldsymbol{\theta}^*\|, \tag{92}$$

for some $0 \leq \alpha < 1$. Here $\mathcal{T}_{\text{loc}}^{\text{s}}(\boldsymbol{\theta}^*)$ is tangent to the hyperplane that contains $\boldsymbol{\theta}^*$ and is spanned by the nonzero eigenvectors of $\nabla^2 \mathcal{L}(\boldsymbol{\theta}^*)$. Thus, assume that the initial point $\boldsymbol{\theta}_0 \in \mathcal{T}_{\text{loc}}^{\text{s}}(\boldsymbol{\theta}^*)$, then

$$\|T(\boldsymbol{\theta}_t) - \boldsymbol{\theta}^*\| \leq \alpha^t \|T(\boldsymbol{\theta}_0) - \boldsymbol{\theta}^*\| \xrightarrow[t \to \infty]{} 0, \tag{93}$$

which shows that with any initialization in $\mathcal{T}_{\text{loc}}^{\text{s}}(\boldsymbol{\theta}^*)$, GD's iterations converge to $\boldsymbol{\theta}^*$.

Next, note that there is a neighborhood of $\boldsymbol{\theta}^*$ within which the set of global minima of the loss form a smooth $m_{\text{Null}}$-dimensional manifold[11]. Let us denote this set of global minima around $\boldsymbol{\theta}^*$ by $\Theta^*$, and set $\eta < 2/\max_{\boldsymbol{\theta} \in \Theta^*}\{\lambda_{\max}(\nabla^2 \mathcal{L})\}$ and limit the set $\Theta^*$ such that $\forall i : \eta \neq 1/\lambda_i$. Then, for each minimum $\boldsymbol{\theta} \in \Theta^*$, according to the first part of the proof, there exists a $(m - m_{\text{Null}})$-dimensional set $\mathcal{T}_{\text{loc}}^{\text{s}}(\boldsymbol{\theta})$. Now, since each $\mathcal{T}_{\text{loc}}^{\text{s}}(\boldsymbol{\theta})$ is contained in a hyperplane that is orthogonal to the tangent of $\Theta^*$ at $\boldsymbol{\theta}$, and the dimension of the tangent of $\Theta^*$ at $\boldsymbol{\theta}^*$ is $m_{\text{Null}}$, we have that the dimension of the union of these sets, $\bigcup_{\boldsymbol{\theta} \in \Theta^*} \mathcal{T}_{\text{loc}}^{\text{s}}(\boldsymbol{\theta})$, is $m$. Thus, the set $\bigcup_{\boldsymbol{\theta} \in \Theta^*} \mathcal{T}_{\text{loc}}^{\text{s}}(\boldsymbol{\theta})$ is of nonzero Lebesgue measure within $\mathbb{R}^m$ and for any initialization in $\bigcup_{\boldsymbol{\theta} \in \Theta^*} \mathcal{T}_{\text{loc}}^{\text{s}}(\boldsymbol{\theta})$, GD converges to $p(\boldsymbol{x})$.

## J  GENERAL LOSS FUNCTIONS

In this section, we discuss how our results can be extended to general loss function with a unique finite root. Assume some general loss function

$$\mathcal{L}(f) = \frac{1}{n} \sum_{j=1}^{n} \ell\left(f(\boldsymbol{x}_j), y_j\right), \tag{94}$$

where $\ell(a, b)$ is twice differentiable w.r.t. $a$ and is minimized when $a = b$, i.e.,

$$\ell'(a, b) \triangleq \frac{\partial}{\partial a} \ell(a, b) = 0 \qquad \forall a = b. \tag{95}$$

Then, we can calculate the loss' gradient

$$\nabla_{\boldsymbol{\theta}} \mathcal{L} = \frac{1}{n} \sum_{j=1}^{n} \ell'\left(f(\boldsymbol{x}_j), y_j\right) \nabla_{\boldsymbol{\theta}} f\left(\boldsymbol{x}_j\right), \tag{96}$$

and Hessian matrix

$$\begin{aligned} \nabla_{\boldsymbol{\theta}}^2 \mathcal{L} &= \frac{1}{n} \sum_{j=1}^{n} \ell''\left(f(\boldsymbol{x}_j), y_j\right) \nabla_{\boldsymbol{\theta}} f\left(\boldsymbol{x}_j\right) \nabla_{\boldsymbol{\theta}} f\left(\boldsymbol{x}_j\right)^{\top} + \frac{1}{n} \sum_{j=1}^{n} \ell'\left(f(\boldsymbol{x}_j), y_j\right) \nabla_{\boldsymbol{\theta}}^2 f\left(\boldsymbol{x}_j\right) \\ &= \frac{1}{n} \sum_{j=1}^{n} \ell''\left(f(\boldsymbol{x}_j), y_j\right) \nabla_{\boldsymbol{\theta}} f\left(\boldsymbol{x}_j\right) \nabla_{\boldsymbol{\theta}} f\left(\boldsymbol{x}_j\right)^{\top}, \end{aligned} \tag{97}$$

where $\ell''(a, b) \triangleq \frac{\partial^2}{\partial a^2} \ell(a, b)$, and in the last transition we used $f(\boldsymbol{x}_j) = y_j$ and $\ell'(a, a) = 0$ for all $a \in \mathbb{R}$. If $\ell''\left(f(\boldsymbol{x}_j), y_j\right) = C > 0$ for all training points, then we can generalize our results by simply multiplying the RHS of (47) by $C$. If not, the analysis can still be used but we need to add a weigtning term to the stability norm which depends on the value of $\ell''\left(f(\boldsymbol{x}_j), y_j\right)$ for each data point.

---

[10]Specifically, any $\eta$ such that $\eta \leq 2/\lambda_{\max}(\nabla^2 \mathcal{L}(\boldsymbol{\theta}^*))$ and $\forall i : \eta \neq 1/\lambda_i$, satisfies this condition.

[11]This set corresponds to multiplying the weights of corresponding neurons within different layers by positive factors whose product is 1.

## K  PROOF OF LEMMA 2

In this section, our goal is to upper bound the top eigenvalue of the flattest implementation of a predictor function $f$. Here we use notation and some derivations form App. F.1. Let $\boldsymbol{q}$ denote the top right singular vector of $\boldsymbol{\Phi}$, then

$$
\begin{aligned}
&\lambda_{\max}\left(\nabla_{\boldsymbol{\theta}}^2 \mathcal{L}\right) \\
&= \frac{1}{n}\left\|\boldsymbol{\Phi}\boldsymbol{q}\right\|^2 \\
&= \frac{1}{n}\left[\left(\sum_{j=1}^{n} q_j\right)^2 + \sum_{i=1}^{k}\left[\sum_{l=1}^{d}\left(\sum_{j=1}^{n} q_j w_i^{(2)} \boldsymbol{x}_{j,l}\mathbb{I}_{j,i}\right)^2 + \left(\sum_{j=1}^{n} q_j w_i^{(2)}\mathbb{I}_{j,i}\right)^2\right.\right. \\
&\hspace{6cm}\left.\left. + \left(\sum_{j=1}^{n} q_j \sigma\left(\boldsymbol{x}_j^\top \boldsymbol{w}_i^{(1)} + b_i^{(1)}\right)\right)^2\right]\right] \\
&\leq \frac{1}{n}\left[n + \sum_{i=1}^{k}\left[\left(w_i^{(2)}\right)^2\left(\sum_{l=1}^{d}\left(\sum_{j=1}^{n} q_j \boldsymbol{x}_{j,l}\mathbb{I}_{j,i}\right)^2 + \left(\sum_{j=1}^{n} q_j\mathbb{I}_{j,i}\right)^2\right)\right.\right. \\
&\hspace{5cm}\left.\left. + \left(\sum_{j=1}^{n} q_j\left\|\boldsymbol{w}_i^{(1)}\right\|\sigma\left(\boldsymbol{x}_j^\top \bar{\boldsymbol{w}}_i^{(1)} - \bar{b}_i^{(1)}\right)\right)^2\right]\right] \quad (98)
\end{aligned}
$$

where in the inequality we used $\left(\sum_{j=1}^{n} u_j\right)^2 \leq n$ for all $\boldsymbol{u} \in \mathbb{S}^{n-1}$ and substituted

$$
\bar{\boldsymbol{w}}_i^{(1)} \triangleq \frac{\boldsymbol{w}_i^{(1)}}{\left\|\boldsymbol{w}_i^{(1)}\right\|}, \qquad \bar{b}_i^{(1)} \triangleq \frac{-b_i^{(1)}}{\left\|\boldsymbol{w}_i^{(1)}\right\|} . \quad (99)
$$

Let $\Theta(f)$ be the set of all implementations corresponding to $f$. Since substituting

$$
w_i^{(1)} \to c_i^{-1} w_i^{(1)} \qquad b_i^{(1)} \to c_i^{-1} b_i^{(1)} \qquad w_i^{(2)} \to c_i w_i^{(2)} \quad (100)
$$

does not affect the network's functionality $f$, we have

$$
\begin{aligned}
&\min_{\boldsymbol{\theta}\in\Theta(f)} \lambda_{\max}\left(\nabla_{\boldsymbol{\theta}}^2 \mathcal{L}\right) \\
&\leq \min_{\boldsymbol{\theta}\in\Theta(f)} \frac{1}{n}\left[n + \sum_{i=1}^{k}\left[\left(w_i^{(2)}\right)^2\left(\sum_{l=1}^{d}\left(\sum_{j=1}^{n} q_j \boldsymbol{x}_{j,l}\mathbb{I}_{j,i}\right)^2 + \left(\sum_{j=1}^{n} q_j\mathbb{I}_{j,i}\right)^2\right)\right.\right. \\
&\hspace{5cm}\left.\left. + \left(\sum_{j=1}^{n} q_j\left\|\boldsymbol{w}_i^{(1)}\right\|\sigma\left(\boldsymbol{x}_j^\top \bar{\boldsymbol{w}}_i^{(1)} - \bar{b}_i^{(1)}\right)\right)^2\right]\right] \\
&= \min_{\boldsymbol{\theta}\in\Theta(f), c_i^2 > 0} \frac{1}{n}\left[n + \sum_{i=1}^{k}\left[c_i^2\left(w_i^{(2)}\right)^2\left(\sum_{l=1}^{d}\left(\sum_{j=1}^{n} q_j \boldsymbol{x}_{j,l}\mathbb{I}_{j,i}\right)^2 + \left(\sum_{j=1}^{n} q_j\mathbb{I}_{j,i}\right)^2\right)\right.\right. \\
&\hspace{4cm}\left.\left. + c_i^{-2}\left\|\boldsymbol{w}_i^{(1)}\right\|^2\left(\sum_{j=1}^{n} q_j \sigma\left(\boldsymbol{x}_j^\top \bar{\boldsymbol{w}}_i^{(1)} - \bar{b}_i^{(1)}\right)\right)^2\right]\right] . \quad (101)
\end{aligned}
$$

A necessary condition for optimality is that the derivative of the objective with respect to $c_i$ is equal to zero:

$$2c_i \left(w_i^{(2)}\right)^2 \left(\sum_{l=1}^{d} \left(\sum_{j=1}^{n} q_j \boldsymbol{x}_{j,l} \mathbb{I}_{j,i}\right)^2 + \left(\sum_{j=1}^{n} q_j \mathbb{I}_{j,i}\right)^2\right)$$

$$- 2c_i^{-3} \left\|\boldsymbol{w}_i^{(1)}\right\|^2 \left(\sum_{j=1}^{n} q_j \sigma \left(\boldsymbol{x}_j^\top \bar{\boldsymbol{w}}_i^{(1)} - \bar{b}_i^{(1)}\right)\right)^2 = 0$$

$$\Rightarrow c_i^2 = \frac{\left\|\boldsymbol{w}_i^{(1)}\right\| \left|\sum_{j=1}^{n} q_j \sigma \left(\boldsymbol{x}_j^\top \bar{\boldsymbol{w}}_i^{(1)} - \bar{b}_i^{(1)}\right)\right|}{\left|w_i^{(2)}\right| \sqrt{\sum_{l=1}^{d} \left(\sum_{j=1}^{n} q_j \boldsymbol{x}_{j,l} \mathbb{I}_{j,i}\right)^2 + \left(\sum_{j=1}^{n} q_j \mathbb{I}_{j,i}\right)^2}}. \tag{102}$$

It is easy to verify that these solutions for $\{c_i\}$ are indeed global minima. Plugging this in, we get

$$\min_{\boldsymbol{\theta} \in \Theta(f)} \lambda_{\max}\left(\nabla_{\boldsymbol{\theta}}^2 \mathcal{L}\right)$$

$$\leq \min_{\boldsymbol{\theta} \in \Theta(f)} \frac{1}{n} \left[n + 2 \sum_{i=1}^{k} \left[\left\|\boldsymbol{w}_i^{(1)}\right\| \left|w_i^{(2)}\right| \left|\sum_{j=1}^{n} q_j \sigma \left(\boldsymbol{x}_j^\top \bar{\boldsymbol{w}}_i^{(1)} - \bar{b}_i^{(1)}\right)\right|\right.\right.$$

$$\left.\left. \times \sqrt{\sum_{l=1}^{d} \left(\sum_{j=1}^{n} q_j \boldsymbol{x}_{j,l} \mathbb{I}_{j,i}\right)^2 + \left(\sum_{j=1}^{n} q_j \mathbb{I}_{j,i}\right)^2}\right]\right]. \tag{103}$$

Now, by Cauchy–Schwarz inequality three times we get

$$\left|\sum_{j=1}^{n} q_j \sigma \left(\boldsymbol{x}_j^\top \bar{\boldsymbol{w}}_i^{(1)} - \bar{b}_i^{(1)}\right)\right| \leq \|\boldsymbol{q}\| \sqrt{\sum_{j=1}^{n} \sigma^2 \left(\boldsymbol{x}_j^\top \bar{\boldsymbol{w}}_i^{(1)} - \bar{b}_i^{(1)}\right)},$$

$$\left(\sum_{j=1}^{n} q_j \boldsymbol{x}_{j,l} \mathbb{I}_{j,i}\right)^2 \leq \|\boldsymbol{q}\|^2 \sum_{j=1}^{n} \boldsymbol{x}_{j,l}^2 \mathbb{I}_{j,i},$$

$$\left(\sum_{j=1}^{n} q_j \mathbb{I}_{j,i}\right)^2 \leq \|\boldsymbol{q}\|^2 \sum_{j=1}^{n} \mathbb{I}_{j,i}. \tag{104}$$

Since $\|\boldsymbol{q}\| = 1$, the right hand sides of these inequalities are independent of $\boldsymbol{q}$. Thus, using these inequalities to further upper bound the top eigenvalue we have

$$\min_{\boldsymbol{\theta} \in \Theta(f)} \lambda_{\max}\left(\nabla_{\boldsymbol{\theta}}^2 \mathcal{L}\right)$$

$$\leq \min_{\boldsymbol{\theta} \in \Theta(f)} \left[1 + \frac{2}{n} \sum_{i=1}^{k} \left[\left\|\boldsymbol{w}_i^{(1)}\right\| \left|w_i^{(2)}\right| \sqrt{\sum_{j=1}^{n} \sigma^2 \left(\boldsymbol{x}_j^\top \bar{\boldsymbol{w}}_i^{(1)} - \bar{b}_i^{(1)}\right)} \sqrt{\sum_{l=1}^{d} \left(\sum_{j=1}^{n} \boldsymbol{x}_{j,l}^2 \mathbb{I}_{j,i}\right) + \left(\sum_{j=1}^{n} \mathbb{I}_{j,i}\right)}\right]\right]$$

$$= 1 + \frac{2}{n} \min_{\boldsymbol{\theta} \in \Theta(f)} \sum_{i=1}^{k} \left\|\boldsymbol{w}_i^{(1)}\right\| \left|w_i^{(2)}\right| \sqrt{\sum_{j=1}^{n} \sigma^2 \left(\boldsymbol{x}_j^\top \bar{\boldsymbol{w}}_i^{(1)} - \bar{b}_i^{(1)}\right)} \sqrt{\sum_{j=1}^{n} \left(\|\boldsymbol{x}_j\|^2 + 1\right) \mathbb{I}_{j,i}}. \tag{105}$$

To continue upper bounding the sharpness $(\lambda_{\max}(\nabla^2 \mathcal{L}))$ of the flattest implementation, we can consider some implementation of $f$. Specifically, since $f \in \mathcal{F}_k$, it can be represented as

$$f(\boldsymbol{x}) = \sum_{i=1}^{k'} a_i \sigma(\boldsymbol{v}_i^\top \boldsymbol{x} - b_i) + \beta \boldsymbol{x}^\top \boldsymbol{h} + c, \tag{106}$$

where $\|\boldsymbol{v}_i\| = 1$ for all $i$, $\|\boldsymbol{h}\| = 1$, $a_i \neq 0$ for all $i$, and $(\boldsymbol{v}_i, b_i) \neq \pm(\boldsymbol{v}_j, b_j)$ for all $i \neq j$ (*i.e.*, the knots of all ReLU units are distinct[12]), and where $k' \leq k$. Thus, we use the following implementation of $f$

$$\boldsymbol{w}_i^{(1)} = \boldsymbol{v}_i, \quad b_i^{(1)} = -b_i, \quad w_i^{(2)} = a_i, \quad b^{(2)} = c + \beta\tau, \tag{107}$$

for $i \in [k']$, where

$$\tau = \frac{1}{n}\sum_{j=1}^{n} \boldsymbol{h}^\top \boldsymbol{x}_j. \tag{108}$$

Additionally, if needed, we add two ReLU neurons to implement the linear component.

$$\boldsymbol{w}_{k'+1}^{(1)} = \boldsymbol{h}, \quad b_{k'+1}^{(1)} = -\tau, \quad w_{k'+1}^{(2)} = \beta,$$
$$\boldsymbol{w}_{k'+2}^{(1)} = -\boldsymbol{h}, \quad b_{k'+2}^{(1)} = \tau, \quad w_{k'+2}^{(2)} = -\beta. \tag{109}$$

Thus,

$$1 + \frac{2}{n}\min_{\boldsymbol{\theta}\in\Theta(f)}\sum_{i=1}^{k}\left\|\boldsymbol{w}_i^{(1)}\right\|\left|w_i^{(2)}\right|\sqrt{\sum_{j=1}^{n}\sigma^2\left(\boldsymbol{x}_j^\top\bar{\boldsymbol{w}}_i^{(1)} - \bar{b}_i^{(1)}\right)}\sqrt{\sum_{j=1}^{n}\left(\|\boldsymbol{x}_j\|^2 + 1\right)\mathbb{I}_{j,i}}$$

$$\leq 1 + \frac{2}{n}\sum_{i=1}^{k'}|a_i|\sqrt{\sum_{j=1}^{n}\sigma^2\left(\boldsymbol{x}_j^\top\boldsymbol{v}_i - b_i\right)}\sqrt{\sum_{j=1}^{n}\left(\|\boldsymbol{x}_j\|^2 + 1\right)\mathbb{I}_{j,i}}$$

$$+ \frac{2}{n}\left(|\beta|\sqrt{\sum_{j=1}^{n}\sigma^2\left(\boldsymbol{x}_j^\top\boldsymbol{h} - \tau\right)}\sqrt{\sum_{j=1}^{n}\left(\|\boldsymbol{x}_j\|^2 + 1\right)\mathbb{I}_{j,k'+1}}\right.$$

$$\left. + |\beta|\sqrt{\sum_{j=1}^{n}\sigma^2\left(-\boldsymbol{x}_j^\top\boldsymbol{h} + \tau\right)}\sqrt{\sum_{j=1}^{n}\left(\|\boldsymbol{x}_j\|^2 + 1\right)\mathbb{I}_{j,k'+2}}\right)$$

$$\leq 1 + \frac{2}{n}\sum_{i=1}^{k'}|a_i|\sqrt{\sum_{j=1}^{n}\sigma^2\left(\boldsymbol{x}_j^\top\boldsymbol{v}_i - b_i\right)}\sqrt{\sum_{j=1}^{n}\left(\|\boldsymbol{x}_j\|^2 + 1\right)\mathbb{I}_{j,i}}$$

$$+ \frac{4|\beta|}{n}\sqrt{\sum_{j=1}^{n}\left(\boldsymbol{x}_j^\top\boldsymbol{h} - \tau\right)^2}\sqrt{\sum_{j=1}^{n}\left(\|\boldsymbol{x}_j\|^2 + 1\right)}, \tag{110}$$

where in the last inequality we used

$$\max\left\{\sigma^2\left(\boldsymbol{x}_j^\top\boldsymbol{h} - \tau\right), \sigma^2\left(-\boldsymbol{x}_j^\top\boldsymbol{h} + \tau\right)\right\} \leq \left(\boldsymbol{x}_j^\top\boldsymbol{h} - \tau\right)^2, \tag{111}$$

and $\|\boldsymbol{x}_j\|^2 + 1 > 0$ and thus removing the indicator term only increases the RHS of (110). Recall that we denoted $C_i \subseteq \{\boldsymbol{x}_j\}$ be the set of training points for which the $i$th neuron is active, and $n_i = |C_i|$. Then,

$$\frac{2}{n}\sum_{i=1}^{k'}|a_i|\sqrt{\sum_{j=1}^{n}\sigma^2\left(\boldsymbol{x}_j^\top\boldsymbol{v}_i - b_i\right)}\sqrt{\sum_{j=1}^{n}\left(\|\boldsymbol{x}_j\|^2 + 1\right)\mathbb{I}_{j,i}}$$

$$= 2\sum_{i=1}^{k'}|a_i|\frac{n_i}{n}\sqrt{\frac{1}{n_i}\sum_{j\in C_i}\left(\boldsymbol{x}_j^\top\boldsymbol{v}_i - b_i\right)^2}\sqrt{\frac{1}{n_i}\sum_{j\in C_i}\left(\|\boldsymbol{x}_j\|^2 + 1\right)}$$

$$= 2\sum_{i=1}^{k'}|a_i|\mathbb{P}\left(\boldsymbol{x}^\top\boldsymbol{v}_i > b_i\right)\sqrt{\mathbb{E}\left[\left(\boldsymbol{x}^\top\boldsymbol{v}_i - b_i\right)^2\Big|\boldsymbol{x}^\top\boldsymbol{v}_i > b_i\right]}\sqrt{\mathbb{E}\left[1 + \|\boldsymbol{x}\|^2\Big|\boldsymbol{x}^\top\boldsymbol{v}_i > b_i\right]}. \tag{112}$$

---

[12]If $(\boldsymbol{v}_i, b_i) = (\boldsymbol{v}_j, b_j)$ then one of the neurons is redundant. If $(\boldsymbol{v}_i, b_i) = -(\boldsymbol{v}_j, b_j)$ then these units can be combined into an affine function, which we "absorb" into the term $\alpha\boldsymbol{x}^\top\boldsymbol{h} + c$.

Additionally,

$$\frac{4|\beta|}{n}\sqrt{\sum_{j=1}^{n}\left(\boldsymbol{x}_j^\top \boldsymbol{h} - \tau\right)^2}\sqrt{\sum_{j=1}^{n}\left(\|\boldsymbol{x}_j\|^2 + 1\right)} = 4|\beta|\sqrt{\mathbb{V}\mathrm{ar}\left(\boldsymbol{x}^\top \boldsymbol{h}\right)}\sqrt{1 + \mathbb{E}\left[\|\boldsymbol{x}\|^2\right]}. \tag{113}$$

Define

$$\hat{g}(\boldsymbol{v}, b) = \mathbb{P}\left(\boldsymbol{x}^\top \boldsymbol{v} > b\right)\sqrt{\mathbb{E}\left[\left(\boldsymbol{x}^\top \boldsymbol{v} - b\right)^2 \Big| \boldsymbol{x}^\top \boldsymbol{v} > b\right]}\sqrt{1 + \mathbb{E}\left[\|\boldsymbol{x}\|^2 \Big| \boldsymbol{x}^\top \boldsymbol{v} > b\right]}. \tag{114}$$

Then, we have

$$\min_{\boldsymbol{\theta} \in \Theta(f)} \lambda_{\max}\left(\nabla_{\boldsymbol{\theta}}^2 \mathcal{L}\right) \le 1 + 2\sum_{i=1}^{k'} |a_i|\hat{g}(\boldsymbol{v}_i, b_i) + 4|\beta|\sqrt{\mathbb{V}\mathrm{ar}\left(\boldsymbol{x}^\top \boldsymbol{h}\right)}\sqrt{1 + \mathbb{E}\left[\|\boldsymbol{x}\|^2\right]}$$

$$= 1 + 2\int_{\mathbb{S}^{d-1}\times\mathbb{R}} \left|\left[(\mathcal{R}^*)^{-1}\Delta f\right](\boldsymbol{v}, b)\right| \hat{g}(\boldsymbol{v}, b)\mathrm{d}s(\boldsymbol{v})\mathrm{d}b$$

$$+ 4|\beta|\sqrt{\mathbb{V}\mathrm{ar}\left(\boldsymbol{x}^\top \boldsymbol{h}\right)}\sqrt{1 + \mathbb{E}\left[\|\boldsymbol{x}\|^2\right]}$$

$$= 1 + 2\|f\|_{\mathcal{R},\hat{g}} + 4|\beta|\sqrt{\mathbb{V}\mathrm{ar}\left(\boldsymbol{x}^\top \boldsymbol{h}\right)}\sqrt{1 + \mathbb{E}\left[\|\boldsymbol{x}\|^2\right]}. \tag{115}$$

Note that,

$$\sqrt{\mathbb{V}\mathrm{ar}\left(\boldsymbol{x}^\top \boldsymbol{h}\right)} \le \sqrt{\lambda_{\max}\left(\boldsymbol{\Sigma}_{\boldsymbol{x}}\right)}, \tag{116}$$

where $\boldsymbol{\Sigma}_{\boldsymbol{x}}$ is the covariance matrix of $\boldsymbol{x}$. Additionally,

$$\|\nabla f(\boldsymbol{x})\| = \left\|\sum_{i=1}^{k'} a_i \mathbf{1}_{\boldsymbol{v}_i^\top \boldsymbol{x} - b_i > 0}\boldsymbol{v}_i + \beta\boldsymbol{h}\right\|$$

$$\ge |\beta|\|\boldsymbol{h}\| - \sum_{i=1}^{k'}\left\|a_i \mathbf{1}_{\boldsymbol{v}_i^\top \boldsymbol{x} - b_i > 0}\boldsymbol{v}_i\right\|$$

$$= |\beta| - \sum_{i=1}^{k'} |a_i|\|\boldsymbol{v}_i\|\mathbf{1}_{\boldsymbol{v}_i^\top \boldsymbol{x} - b_i > 0}$$

$$\ge |\beta| - \sum_{i=1}^{k'} |a_i|$$

$$= |\beta| - \int_{\mathbb{S}^{d-1}\times\mathbb{R}}\left|(\mathcal{R}^*)^{-1}\Delta f\right|\mathrm{d}s(\boldsymbol{v})\mathrm{d}b$$

$$= |\beta| - \|f\|_{\mathcal{R}}. \tag{117}$$

Therefore, for any $\boldsymbol{x} \in \mathbb{R}^d$

$$|\beta| \le \|\nabla f(\boldsymbol{x})\| + \|f\|_{\mathcal{R}}. \tag{118}$$

Taking the tightest bound we obtain

$$|\beta| \le \|f\|_{\mathcal{R}} + \inf_{\boldsymbol{x} \in \mathbb{R}^d}\|\nabla f(\boldsymbol{x})\|. \tag{119}$$

Overall, combining (115), (116), and (119) we obtain

$$\min_{\boldsymbol{\theta} \in \Theta(f)} \lambda_{\max}\left(\nabla_{\boldsymbol{\theta}}^2 \mathcal{L}\right) \le 1 + 2\|f\|_{\mathcal{R},\hat{g}} + 4\left(\|f\|_{\mathcal{R}} + \inf_{\boldsymbol{x} \in \mathbb{R}^d}\|\nabla f(\boldsymbol{x})\|\right)\sqrt{\lambda_{\max}\left(\boldsymbol{\Sigma}_{\boldsymbol{x}}\right)}\sqrt{1 + \mathbb{E}\left[\|\boldsymbol{x}\|^2\right]}. \tag{120}$$

## L  PROOF OF PROPOSITION 3

Let $f \in W_w^{d+1,1}(\mathbb{R}^d)$. First, we show that this implies both $\|f\|_{\mathcal{R}}$ and $\|f\|_{\mathcal{R},\hat{g}}$ are finite. Since we assume $d$ is odd $(-\Delta)^{(d+1)/2}f$ is an integral power of the negative Laplacian applied to $f$, hence can be expanded as a linear combination of $d+1$ order partial derivatives, and so

$$\|(-\Delta)^{(d+1)/2}f\|_{1,w} \leq a_d \sum_{|\beta|=d+1} \|\partial^\beta f\|_{1,w} \leq a_d \|f\|_{W_w^{d+1,1}(\mathbb{R}^d)}, \tag{121}$$

where $a_d$ is a constant depending on $d$ but independent of $f$. Therefore, $\|(-\Delta)^{(d+1)/2}f\|_{1,w}$ is finite. In particular, this shows $(-\Delta)^{(d+1)/2} \in L^1(\mathbb{R}^d)$, and so $\mathcal{R}(-\Delta)^{(d+1)/2}f$ exists in a classical sense. This implies have the formulas $\|f\|_{\mathcal{R}} = \gamma_d\|\mathcal{R}(-\Delta)^{(d+1)/2}f\|_1$ and $\|f\|_{\mathcal{R},\hat{g}} = \gamma_d\|\hat{g} \cdot \mathcal{R}(-\Delta)^{(d+1)/2}f\|_1$ where $\gamma_d = \frac{1}{2(2\pi)^{d-1}}$ (see (Ongie et al., 2020, Prop. 1)).

Recall that $w(\boldsymbol{x}) = \mathcal{R}^*[1+|b|](\boldsymbol{x}) = c_d + \zeta_d\|\boldsymbol{x}\|$, with $c_d = \int_{\mathbb{S}^{d-1}} d\boldsymbol{v}$ and $\zeta_d = \int_{\mathbb{S}^{d-1}} |v_1| d\boldsymbol{v}$. Therefore we have that

$$
\begin{aligned}
\|(-\Delta)^{(d+1)/2}f\|_{1,w} &= \int_{\mathbb{R}^d} \left|(-\Delta)^{(d+1)/2}f(\boldsymbol{x})\right| w(\boldsymbol{x}) \mathrm{d}\boldsymbol{x} \\
&= \int_{\mathbb{S}^{d-1}\times\mathbb{R}} \mathcal{R}\left\{\left|(-\Delta)^{(d+1)/2}f\right|\right\}(\boldsymbol{v},b)\,(1+|b|)\mathrm{d}s(\boldsymbol{v})\mathrm{d}b \\
&\geq \int_{\mathbb{S}^{d-1}\times\mathbb{R}} \left|\mathcal{R}\left\{(-\Delta)^{(d+1)/2}f\right\}(\boldsymbol{v},b)\right|\,(1+|b|)\mathrm{d}s(\boldsymbol{v})\mathrm{d}b \\
&= \int_{\mathbb{S}^{d-1}\times\mathbb{R}} \left|\mathcal{R}\left\{(-\Delta)^{(d+1)/2}f\right\}\right| \frac{1+|b|}{1+\hat{g}(\boldsymbol{v},b)}(1+\hat{g}(\boldsymbol{v},b))\mathrm{d}s(\boldsymbol{v})\mathrm{d}b \\
&\geq C_{\hat{g}} \int_{\mathbb{S}^{d-1}\times\mathbb{R}} \left|\mathcal{R}\left\{(-\Delta)^{(d+1)/2}f\right\}\right|(1+\hat{g}(\boldsymbol{v},b))\mathrm{d}s(\boldsymbol{v})\mathrm{d}b \\
&= \gamma_d^{-1}C_{\hat{g}}(\|f\|_{\mathcal{R}} + \|f\|_{\mathcal{R},\hat{g}}), \tag{122}
\end{aligned}
$$

where $C_{\hat{g}} = \inf_{(\boldsymbol{v},b)\in\mathbb{S}^{d-1}\times\mathbb{R}} \frac{1+|b|}{1+\hat{g}(\boldsymbol{v},b)}$ is finite and non-zero because for all $\boldsymbol{v} \in \mathbb{S}^{d-1}$ we have $\hat{g}(\boldsymbol{v},b) = O(|b|)$ as $|b| \to \infty$ where the implied constant is independent of $\boldsymbol{v}$. Therefore, we have shown $\|f\|_{\mathcal{R}}$ and $\|f\|_{\mathcal{R},\hat{g}}$ are finite as claimed.

Let $\alpha = (\mathcal{R}^*)^{-1}\Delta f = -\gamma_d \mathcal{R}(-\Delta)^{(d+1)/2}f$. Then $\|f\|_{\mathcal{R}} = \|\alpha\|_1$ is finite, and so $\alpha$ is an $L^1$ function, which can be identified with a finite signed measure. Since $\|f\|_{\mathcal{R},g} = \|\hat{g}\cdot\alpha\|_1$ is also finite, we see that $\tilde{\alpha}(\boldsymbol{v},b) := (1+\hat{g}(\boldsymbol{v},b))\alpha(\boldsymbol{v},b)$ is also an $L^1$ function which can be identified with a finite signed measure. By (Malliavin et al., 1995, Thm. 6.9), this implies there exists a sequence of finite atomic measures $\{\tilde{\alpha}_k\}$, such that each $\tilde{\alpha}_k$ consists of a sum of at most $k$ Diracs, converging narrowly[13] to $\tilde{\alpha}$ with $\|\tilde{\alpha}_k\|_{\mathrm{TV}} \leq \|\alpha\|_1$. Define $\alpha_k(\boldsymbol{v},b) = \tilde{\alpha}_k(\boldsymbol{v},b)/(1+\hat{g}(\boldsymbol{v},b))$, which is also an atomic measure. Then it is easy to show $\alpha_k \to \alpha$ narrowly, as well. By Lemma 5 of (Ongie et al., 2020), this implies there exists a sequence of single hidden-layer ReLU networks $f_k \in \mathcal{F}_k$ converging to $f$ pointwise. Therefore, for all $k$ we have

$$\|f_k\|_{\mathcal{R}} + \|f_k\|_{\mathcal{R},\hat{g}} = \|f_k\|_{\mathcal{R},1+\hat{g}} = \|\tilde{\alpha}_k\|_{\mathrm{TV}} \leq \|\tilde{\alpha}\|_1 = \|f\|_{\mathcal{R},1+\hat{g}} = \|f\|_{\mathcal{R}} + \|f\|_{\mathcal{R},\hat{g}}. \tag{123}$$

Combining this inequality with the bound on $\|f\|_{\mathcal{R}} + \|f\|_{\mathcal{R},\hat{g}}$ given above, we see that

$$\|f_k\|_{\mathcal{R}} + \|f_k\|_{\mathcal{R},\hat{g}} \leq \|f\|_{\mathcal{R}} + \|f\|_{\mathcal{R},\hat{g}} \leq c_{d,\hat{g}}\|f\|_{W_w^{d+1,1}(\mathbb{R}^d)}, \tag{124}$$

where $c_{d,\hat{g}} = a_d\gamma_d C_{\hat{g}}^{-1}$ is a constant defined independently of $f$.

Finally, if $K$ is any compact subset, the pointwise convergence of $f_k$ to $f$ on $K$ can be upgraded to $L^1$-convergence using Lebesgue's dominated convergence theorem: by the bounds on the Lipschitz constant of a function given in terms of the $\mathcal{R}$-norm in Proposition 8 of (Ongie et al., 2020), we have $|f_k(\boldsymbol{x})| \leq \|\boldsymbol{x}\|(C + \|f_k\|_{\mathcal{R}}) \leq B\|\boldsymbol{x}\|$ for some constants $C, B \geq 0$, and since $\boldsymbol{x} \mapsto B\|\boldsymbol{x}\|$ is $L^1$-integrable over any compact subset, the hypotheses of Lesbesgue's dominated convergence theorem hold.

---

[13]Namely, $\langle\alpha_k,\varphi\rangle \to \langle\alpha,\varphi\rangle$ for all continuous and bounded functions $\varphi : \mathbb{S}^{d-1} \times \mathbb{R} \to \mathbb{R}$.

## M    STABILITY NORM OF "PYRAMID" FUNCTION

Here, to provide a better understanding of the depth separation result in Proposition 2, we show by direct calculation that the "pyramid" function in $d = 2$ dimensions, given by

$$p(\boldsymbol{x}) = p(x_1, x_2) = [1 - |x_1| - |x_2|]_+, \tag{125}$$

fails to have finite stability norm. In particular, we explicitly compute $(\mathcal{R}^*)^{-1}\Delta p$ as a tempered distribution and show it is not a finite measure (*i.e.*, must be a distribution of order $> 0$), which implies it cannot have finite stability norm under the assumptions in Proposition 2.

First, observe that $\Delta p$ is linear combination of Diracs supported on finite line segments $\ell_k$ defining the "edges" of the pyramid:

$$\Delta p(\boldsymbol{x}) = \sum_k c_k \delta_{\ell_k}. \tag{126}$$

This means that if $\phi$ is any Schwartz class test function, then

$$\langle \Delta p(\boldsymbol{x}), \phi \rangle = \sum_k c_k \int_{\ell_k} \phi(\boldsymbol{x})\mathrm{d}s(\boldsymbol{x}). \tag{127}$$

Note that $\Delta p(\boldsymbol{x})$ is a finite measure (*i.e.*, a distribution of order zero), since

$$|\langle \Delta p(\boldsymbol{x}), \phi \rangle| \leq \left( \sum_k |c_k| |\ell_k| \right) \|\phi\|_\infty, \tag{128}$$

where $|\ell_k|$ is the length of the line segment $\ell_k$. See Fig. 6 below for illustration.

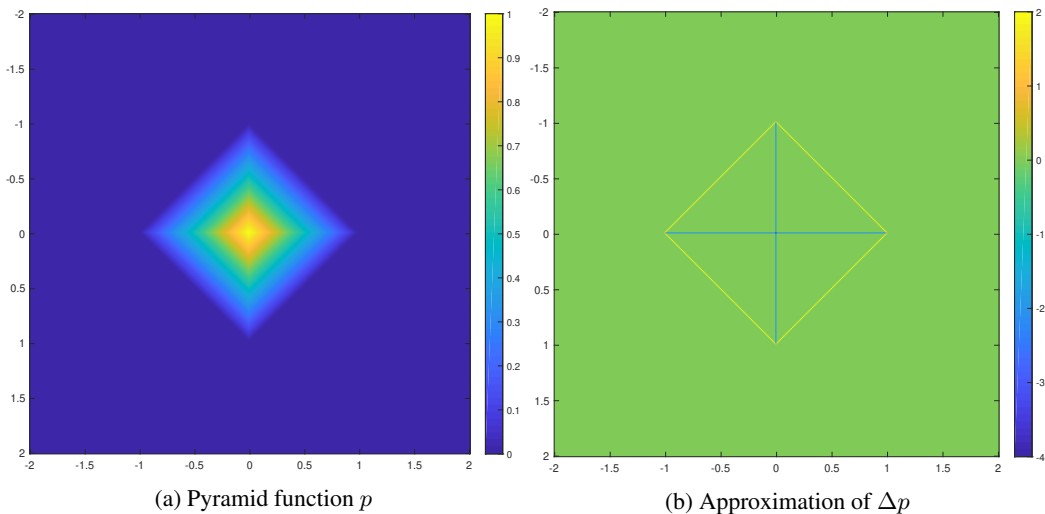

(a) Pyramid function $p$                                     (b) Approximation of $\Delta p$

Figure 6: **Visualizations of the pyramid function $p$ and its Laplacian - $\Delta p$.**

Now we compute $\mathcal{R}\Delta p$, the Radon transform of $\Delta p$, which exists as a tempered distribution. First, we show how to compute $\mathcal{R}\delta_\ell$ where $\ell$ denotes a general line segment. Let $\varphi$ is any Schwartz class test function defined in Radon domain, then

$$\langle \mathcal{R}\delta_\ell, \varphi \rangle = \int_\ell (\mathcal{R}^*\varphi)(\boldsymbol{x})\mathrm{d}s(\boldsymbol{x}) = \int_\ell \int_{\mathbb{S}^1} \varphi(\boldsymbol{v}, \boldsymbol{v}^\top \boldsymbol{x})\mathrm{d}\boldsymbol{v}\mathrm{d}s(\boldsymbol{x}) = \int_{\mathbb{S}^1} \int_\ell \varphi(\boldsymbol{v}, \boldsymbol{v}^\top \boldsymbol{x})\mathrm{d}s(\boldsymbol{x})\mathrm{d}\boldsymbol{v}, \tag{129}$$

where the exchange of integrals is justified by Fubini's theorem since $\delta_\ell$ is a finite measure.

Suppose $\ell$ is a vertical line segment $\ell = \{(0, t) : t \in [c, d]\}$. Assuming $\boldsymbol{v}$ is such that $v_2 \neq 0$, then the inner integral above with $\ell$ in place of $\ell_k$ above simplifies as

$$
\begin{aligned}
\int_\ell \varphi(\boldsymbol{v}, \boldsymbol{v}^\top \boldsymbol{x}) \mathrm{d}s(\boldsymbol{x}) &= \int_c^d \varphi(\boldsymbol{v}, v_2 t) \mathrm{d}t \\
&= \frac{1}{|v_2|} \int_{|v_2|c}^{|v_2|d} \varphi(\boldsymbol{v}, b) \mathrm{d}b \\
&= \int \frac{1}{|v_2|} \mathbb{1}_{[|v_2|a, |v_2|b]}(b) \varphi(\boldsymbol{v}, b) \mathrm{d}b.
\end{aligned}
\tag{130}
$$

In the event that $v_2 = 0$ we have

$$
\int_\ell \phi(\boldsymbol{v}, x_1) \mathrm{d}s(\boldsymbol{x}) = \int_c^d \phi(\boldsymbol{v}, 0) \mathrm{d}t = |d - c| \phi_{\boldsymbol{v}}(0) = |d - c| \langle \delta_0, \phi_{\boldsymbol{v}} \rangle.
\tag{131}
$$

Therefore, we have shown

$$
\mathcal{R}\delta_\ell(\boldsymbol{v}, b) = \begin{cases} \frac{1}{|v_2|} \mathbb{1}_{[|v_2|c, |v_2|d]}(b) & \text{if } v_2 \neq 0, \\ |d - c| \delta(b) & \text{if } v_2 = 0. \end{cases}
\tag{132}
$$

Now, consider one of line segments $\ell_k$ coinciding with the edges of the pyramid. This can be parameterized as $\ell_k = \{b_k \boldsymbol{v}_k + t \boldsymbol{v}_k^\perp : t \in [c_k, d_k]\}$ where $\boldsymbol{v}_k \in \mathbb{S}^1$ is a unit vector, $b_k \in \mathbb{R}$ is a constant, and $\boldsymbol{v}_k^\perp$ is orthogonal to $\boldsymbol{v}_k$. Let $\theta_k$ is the angle such that $\boldsymbol{v}_k = [\cos(\theta_k), \sin(\theta_k)]$, then $\ell_k$ is a rotation of the vertical line segment $\ell$ through the angle $\theta_k$, and translation by $b_k \boldsymbol{v}_k$. Therefore, by properties of Radon transforms,

$$
\mathcal{R}\delta_{\ell_k}(\boldsymbol{v}(\theta), b) = \mathcal{R}\delta_\ell(\boldsymbol{v}(\theta - \theta_k), b - b_k \cos(\theta - \theta_k)),
\tag{133}
$$

where we set $\boldsymbol{v}(\theta) = [\cos(\theta), \sin(\theta)]$ for all $\theta \in [0, \pi)$. More concretely, we can express every slice $\mathcal{R}\delta_{\ell_k}(\boldsymbol{v}, \cdot)$ as either a weighted indicator function when $\boldsymbol{v} \neq \pm\boldsymbol{v}_k$, which is non-zero when $b$ is such that the line $L_{\boldsymbol{v}, b} := \{\boldsymbol{x} : \boldsymbol{v}^\top \boldsymbol{x} = b\}$ intersects the line segment $\ell_k$, or as a weighted Dirac when $\boldsymbol{v} = \pm\boldsymbol{v}_k$, i.e.,

$$
\mathcal{R}\delta_{\ell_k}(\boldsymbol{v}(\theta), b) = \begin{cases} |\sin(\theta - \theta_k)|^{-1} & \text{if } L_{\boldsymbol{v}(\theta), b} \cap \ell_k \text{ is a singleton}, \\ |\ell_k| \delta(b - b_k) & \text{if } \boldsymbol{v} \text{ parallel to } \boldsymbol{v}_k, \\ 0 & \text{else}. \end{cases}
\tag{134}
$$

For a fixed $\boldsymbol{v}(\theta) \in \mathbb{S}^{d-1}$ the set of $b \in \mathbb{R}$ for which $L_{\boldsymbol{v}(\theta), b} \cap \ell_0 \neq \emptyset$ is always a closed interval $[\alpha_\theta, \beta_\theta]$. Therefore, for $\theta \neq \theta_k$ we can write $\mathcal{R}\delta_{\ell_k}(\theta, \cdot) = |\sin(\theta - \theta_k)|^{-1} \mathbb{1}_{[\alpha_k(\theta), \beta_k(\theta)]}$ for some $\alpha_k(\theta)$ and $\beta_k(\theta)$ that vary continuously with $\theta$.

Finally, by linearity, we obtain $\mathcal{R}\Delta p = \sum_k c_k \mathcal{R}\delta_{\ell_k}$. See Figure 7 for an approximate plot of $\mathcal{R}\Delta p$.

Now we compute $(\mathcal{R}^*)^{-1}\Delta p$. Recall that $(\mathcal{R}^*)^{-1} = \mathcal{K}\mathcal{R}$ where $\mathcal{K} = \mathcal{H}\partial_b$ is a filtering step with $\mathcal{H}$ being the Hilbert transform applied separably in the $b$-variable (Helgason, 1999). For a smooth function $g$,

$$
\mathcal{H}g(b) = \frac{1}{\pi} \text{p.v.} \int_{-\infty}^{\infty} \frac{g(b')}{b - b'} \mathrm{d}b',
\tag{135}
$$

where p.v. indicates a principle value integral. Therefore, for any $\theta \neq \theta_k$, we have

$$
\begin{aligned}
\mathcal{K}\mathcal{R}\delta_{\ell_k}(\boldsymbol{v}(\theta), b) &= \frac{1}{|\sin(\theta - \theta_k)|} (\mathcal{H}\delta_{\alpha_k(\theta)} - \mathcal{H}\delta_{\beta_k(\theta)}) \\
&= \frac{1}{\pi |\sin(\theta - \theta_k)|} \left( \text{p.v.} \frac{1}{b - \alpha_k(\theta)} - \text{p.v.} \frac{1}{b - \beta_k(\theta)} \right),
\end{aligned}
\tag{136}
$$

and for $\theta = \theta_k$ we have

$$
\mathcal{K}\mathcal{R}\delta_{\ell_k}(\boldsymbol{v}(\theta_k), b) = |\ell_k| \mathcal{H}\delta'(b - b_k) = -\frac{|\ell_k|}{\pi} \text{p.v.} \frac{1}{(b - b_k)^2}.
\tag{137}
$$

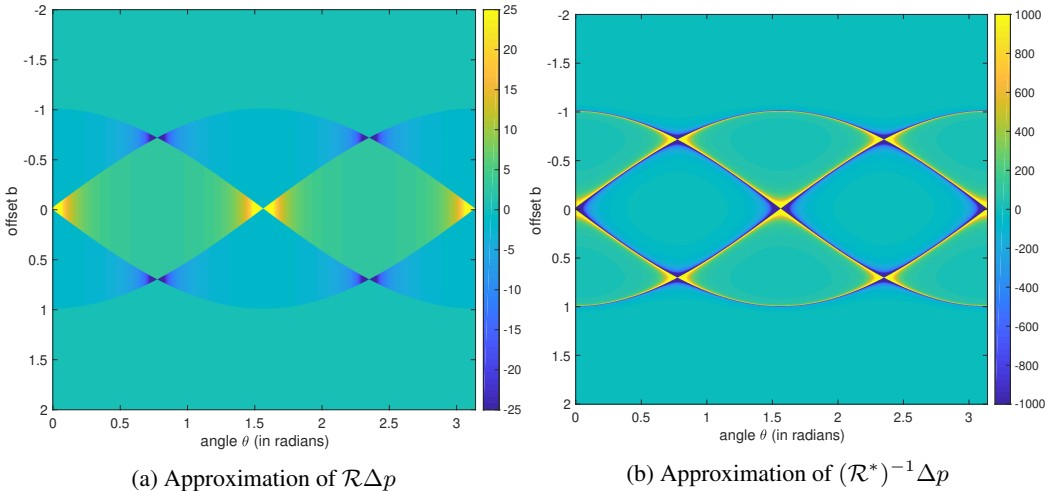

(a) Approximation of $\mathcal{R}\Delta p$       (b) Approximation of $(\mathcal{R}^*)^{-1}\Delta p$

Figure 7: **Visualizations of $\mathcal{R}\Delta p$ and $(\mathcal{R}^*)^{-1}\Delta p$.**

Finally, by linearity of the operator $\mathcal{KR}$, we have

$$(\mathcal{R}^*)^{-1}\Delta p = \mathcal{KR}\Delta p = \sum_k c_k \mathcal{KR}\delta_{\ell_k}. \tag{138}$$

Thus,

$$[(\mathcal{R}^*)^{-1}\Delta p](\boldsymbol{v}(\theta), b) = \sum_k c_k |\sin(\theta - \theta_k)|^{-1} \left( \text{p.v.} \frac{1}{b - \alpha_k(\theta)} - \text{p.v.} \frac{1}{b - \beta_k(\theta)} \right)$$
$$+ \sum_k c_k |\ell_k| \delta(\theta - \theta_k) \cdot \text{p.v.} \frac{-1}{(b - b_k)^2}. \tag{139}$$

See Figure 7 for an approximate plot of $(\mathcal{R}^*)^{-1}\Delta p = \mathcal{KR}\Delta p$. As evidenced by the plot, this density has singularities along a 1-D manifold $S$ in Radon domain. This set corresponds to all lines in the primal domain passing through the corners of the pyramid.

Finally, we show that $\alpha := (\mathcal{R}^*)^{-1}\Delta p$ is not a finite measure (*i.e.*, it is not an order zero distribution). Intuitively, this is because the "density" $\alpha(\boldsymbol{v}, b)$ is not absolutely integrable, since every 1-D angular slice has singularities like $1/|b|$. Below we prove this more formally.

To prove $\alpha$ cannot be an order zero distribution, we construct a family of uniformly bounded test functions $\{\varphi_\epsilon\}_{\epsilon > 0}$ such that $|\langle \alpha, \varphi_\epsilon \rangle| \geq \rho(\epsilon) \|\varphi_\epsilon\|_\infty$, where $\rho(\epsilon)$ is a function such that $\rho(\epsilon) \to +\infty$ as $\epsilon \to 0^+$.

Let $\gamma > 0$ be a small fixed constant less than one. For every $0 < \epsilon < \gamma$, consider the "rainbow-shaped" subset of Radon domain $\Omega_\epsilon$ defined by the inequalities $-\gamma/2 < \theta < \gamma/2$ and $\cos(\theta) - \epsilon < b < \cos(\theta)$ where . In primal domain, the set corresponds to a collection of lines that nearly intersect the corner point $(1, 0)$.

Only three terms in the sum making up $(\mathcal{R}^*)^{-1}\Delta p$ in (139) are dominant in the region $\Omega_\epsilon$, corresponding to the three line segments in the support of $\Delta p$ that arise from the right-most corner of the pyramid. Elementary calculations show these three terms are specified by the parameters:

$$c_1 = -2, \;\; \theta_1 = \pi/2, \;\; \beta_1(\theta) = \cos(\theta),$$
$$c_2 = \sqrt{2}, \;\; \theta_2 = \pi/4, \;\; \beta_2(\theta) = \cos(\theta),$$
$$c_3 = \sqrt{2}, \;\; \theta_3 = -\pi/4, \;\; \beta_3(\theta) = \cos(\theta). \tag{140}$$

Therefore, $\alpha$ is well-approximated on $\Omega_\epsilon$ by

$$\tilde{\alpha} = \left( \frac{\sqrt{2}}{|\sin(\theta - \pi/4)|} + \frac{\sqrt{2}}{|\sin(\theta + \pi/4)|} - \frac{2}{|\sin(\theta - \pi/2)|} \right) \text{p.v.} \frac{1}{\cos(\theta) - b}, \tag{141}$$

where we omit the terms p.v. $\frac{-1}{b-\alpha_k(\theta)}$ and p.v. $\frac{-1}{(b-b_k)^2}$, since points in $\Omega_\epsilon$ are far from their singularity set. In particular, we can show $\alpha - \tilde{\alpha}$ is an order zero distribution when restricted to $\Omega_\epsilon$ (*i.e.*, all other terms are locally smooth and bounded). Let $g(\theta)$ be the function of $\theta$ in front of the principle value in $\tilde{\alpha}$. Note that $g(\theta) > B > 0$ for all $\theta \in \Omega_\epsilon$ where the constant $B$ is independent of $\epsilon$.

Let $\varphi_\epsilon(\theta, b)$ be a smooth function supported in $\Omega_\epsilon$ such that $0 \leq \varphi_\epsilon(\theta, b) \leq 1$ and $\varphi_\epsilon(\theta, b) = 1$ on the region defined by the inequalities $-\gamma/2 < \theta < \gamma/2$ and $\cos(\theta) - \epsilon \leq b \leq \cos(\theta) - \epsilon^2$. Then for any fixed $\theta \in (-\gamma/2, \gamma/2)$, the integral p.v. $\int \frac{\varphi_\epsilon(\theta, \cdot)}{\cos(\theta) - b} \mathrm{d}b$ is bounded below by $\int_{\epsilon^2}^{\epsilon} \frac{1}{b} \mathrm{d}b = \log(\epsilon^{-1})$. Therefore, we have

$$
\begin{aligned}
|\langle \alpha, \varphi_\epsilon \rangle| &\geq |\langle \tilde{\alpha}, \varphi_\epsilon \rangle| - |\langle \tilde{\alpha} - \alpha, \varphi_\epsilon \rangle| \\
&\geq \left| \int_{-\gamma/2}^{\gamma/2} \langle \tilde{\alpha}_\theta, \varphi_\epsilon(\theta, \cdot) \rangle d\theta \right| - C \|\varphi_\epsilon\|_\infty \\
&\geq (\gamma B \log(\epsilon^{-1}) - C) \|\varphi_\epsilon\|_\infty.
\end{aligned}
\tag{142}
$$

Since $\|\varphi_\epsilon\|_\infty = 1$ and $\gamma B \log(\epsilon^{-1}) - C \to +\infty$ as $\epsilon \to 0^+$, this shows $\alpha$ cannot be a distribution of order zero, *i.e.*, it cannot be identified with a finite measure.

# N    ADDITIONAL EXPERIMENTS

The experiments in Sec. 6 are designed to demonstrate Theorem 1 in a diverse range of step sizes ($[10^{-4}, 0.1]$). Since flat minima of the loss landscape are concentrated near the origin in parameter space, and training with small step size near flat minima is inefficient, we used large initialization (about 10 times larger than standard methods). Here we repeat the MNIST experiment using various initialization scales on a higher range of step sizes ($[10^{-3}, 0.2]$). Figure 8 presents the sharpness curves for the different scales. For large initialization, $\times 10$ and $\times 15$, we get the same behavior as depicted in Sec. 6. For small initialization, $\times 1$ and $\times 5$, the sharpness of the obtained solutions is fixed for small learning rates up to a critical step size $\eta^*$. At this threshold, the sharpness equals $2/\eta^*$, and any increment in the step size makes the minimum unstable. This pushes SGD to flatter minima for larger step sizes, ones that satisfy the stability criterion. Here it is important to note that for standard initialization, shown in Fig. 8(a), the threshold is well before the standard step size of $\eta = 0.1$. Namely, this phenomenon happens using standard initialization and standard learning rate.

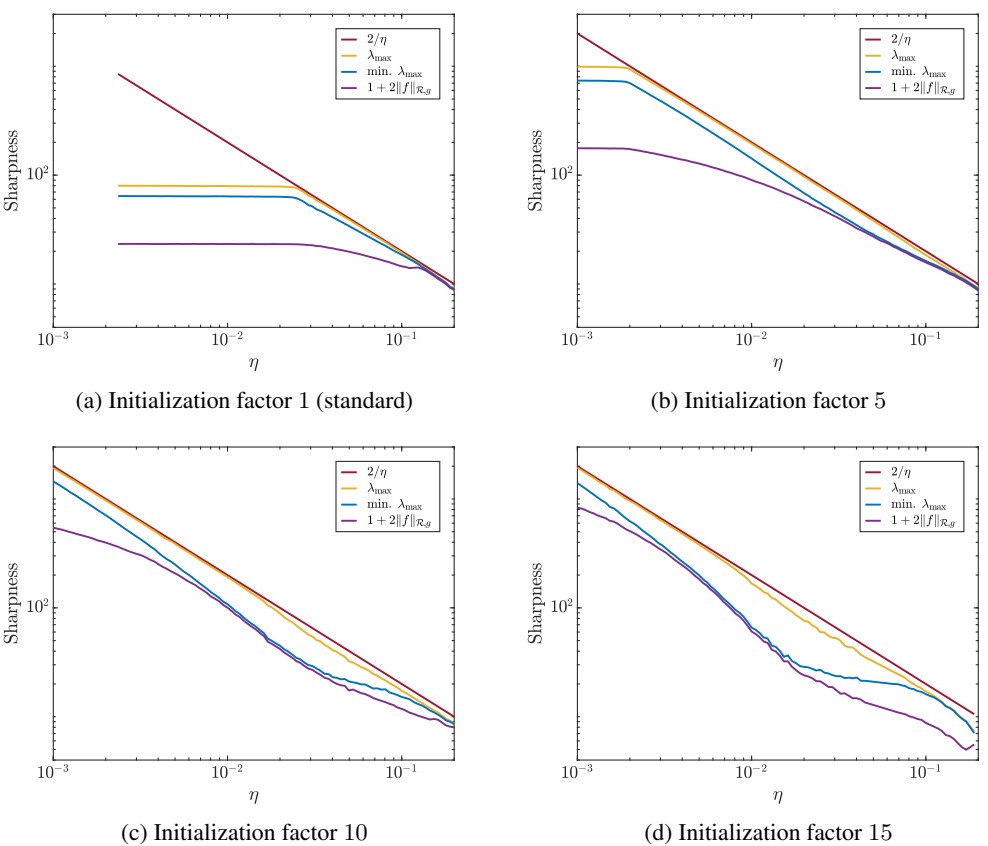

Figure 8: **Sharpness vs. step size for different initialization scales.** We trained a single hidden-layer ReLU network for binary classification on two classes from MNIST using SGD (see Sec. 6 for details). Specifically, we initialized the network using different scales, and for each scale we trained the network using multiple step sizes. We see that as $\eta$ increases, the minima get flatter in parameter space (yellow curve), which translates to smoother predictors in function space (purple curve).

