# OpenReview forum: "The Implicit Bias of Minima Stability in Multivariate Shallow ReLU Networks"
_ICLR.cc/2023/Conference — ICLR 2023 poster_

### Official Review · Reviewer_iAv3 · 2022-10-18

**Confidence:** 4
**Correctness:** 4
**Technical Novelty And Significance:** 3
**Empirical Novelty And Significance:** 3
**Recommendation:** 6

**Clarity, Quality, Novelty And Reproducibility:**

The presentation is clear and precise.

While the results of this paper are new, the proof techniques seem to mostly be a combination of (Mulayoff et al., 2021) and (Ongie et al., 2021). The results in this paper are weaker: in (Ongie et al., 2021) it is shown that the representation equals some form of Random trasform based norm, while in this paper the bound on the stability norm is only a necessary condition (do the authors believe that it is also sufficient? In that case what is missing?).

**Strength And Weaknesses:**

I think that the question of minima stability is very interesting and crucial to understand what type of functions are learned by DNNs. The analysis proposed here seems quite complete for shallow ReLU networks, though it requires a number of technical assumptions and it gives only a necessary condition for stability.

The depth separability example function is both surprisingly simple and interesting. Tt suggests that stability conditions in deeper networks might be very different from those of shallow networks.

**Summary Of The Paper:**

The paper gives a necessary condition on a function $f$ for the existence of a local minimum (of a shallow ReLU network which represents $f$) that is stable with a fixed learning rate $\eta$. The stability condition on $f$ requires a stability norm of $f$ to be bounded. This norm can be interpreted as a form of weighted L1 norm of the Laplacian of $f$.

The authors then give a depth separation result: an example function $f$ which has infinite stability norm (and hance cannot be approximated with a fixed learning rate in shallow ReLU networks), but such that there is a deeper network representing $f$ in stable manner.

Finally the authors show (under some conditions) that any function $f$ in a weighted Sobolev space can be approximated by networks of increasing width $k$ in a stable manner.

**Summary Of The Review:**

The paper gives a rather complete answer to the question of minima stability in shallow ReLU networks, using a combination of techniques from two previous papers.

---

> ### Author Response · Authors · 2022-11-14
> **Author response**
>
> We thank the reviewer for the constructive feedback.
>
> &nbsp;
>
> **Is the necessary bound on the stability norm also sufficient?** \
> This is a great question! Our bound on the stability norm,  *i.e.* Theorem 1, is only necessary since the stability norm only bounds the top eigenvalue of the Hessian and is not equal to it. However, all the ingredients for constructing a sufficient condition for stability in the full batch case (GD) already exist in our paper. Specifically, Lemma 2 states that for any solution $ f \\in \\mathcal{F} $ there exists a corresponding implementation $  \\boldsymbol{\\theta} $ for which
>
> $ \\lambda\_{\\max} (\\nabla^2\_\\boldsymbol{\\theta} \\mathcal{L} )
> \\leq 1+2 \\Vert f \\Vert\_{ \\mathcal{R}, \\hat{g}} + 4 \\left( {\\Vert f \\Vert }\_{\\mathcal{R}} +\\inf\_{\\boldsymbol{x} \\in \mathbb{R}^d }
>  {\\Vert \\nabla f (\\boldsymbol{X}) \\Vert } \\right)  \\sqrt{\\lambda\_{\\max} \\big( \\boldsymbol{\\Sigma}\_{\\boldsymbol{X}} \\big) } \sqrt{ 1+  \\mathbb{E} \\left[ { \\Vert \\boldsymbol{X} \\Vert }^2 \\right]}, $
>
> where  $ \\hat{g} $  is a weight function different from $ g $ (see Lemma 2). It is well known that $ \\lambda\_{\\max} \\leq 2/\\eta $ is a necessary and sufficient condition for linear stability for GD (not SGD). Therefore, we have the sufficient condition (informal):
>
> Let $ f $ be a solution.  Assume that the knots of $ f $ do not coincide with any training point. If
>
> $ \\Vert f \\Vert\_{ \\mathcal{R}, \\hat{g}}  + 2 \\left( {\\Vert f \\Vert }\_{\\mathcal{R}} +\\inf\_{\\boldsymbol{x} \\in \mathbb{R}^d }
>  {\\Vert \\nabla f (\\boldsymbol{X}) \\Vert } \\right)  \\sqrt{\\lambda\_{\\max} \\big( \\boldsymbol{\\Sigma}\_{\\boldsymbol{X}} \\big) } \\sqrt{ 1+  \\mathbb{E} \\left[ { \\Vert \\boldsymbol{X} \\Vert }^2 \\right]} \\leq 1/\\eta - 1/2, $
>
> then $ f $  is a linearly stable solution for GD with step size $ \\eta $.
>
> We will add this result to our paper, thanks!

---

### Official Review · Reviewer_EjYW · 2022-10-19

**Confidence:** 5
**Correctness:** 4
**Technical Novelty And Significance:** 3
**Empirical Novelty And Significance:** Not applicable
**Recommendation:** 8

**Clarity, Quality, Novelty And Reproducibility:**

The writing is clear, the claim is supported sufficiently, and the result is novel.

**Strength And Weaknesses:**

### Strong point

The paper is overall well-written and I can follow most results smoothly.
The theoretical result suggests that the stability of SGD implicitly regularizes the smoothness of learned model whereby SGD with large learning rate  generalizes well. Preivous similar works all are limited to linear networks and univariate two-layer neural networks. This work considers the standard two-layer neural networks, which is much more interesting. Given recent progresses in understanding the interplay between dynamical stability and implicit bias of SGD, I think this work is of great importance.

### Weak point

1. (Ma et al., 2021) also established the connection between stability and the input smoothness of implemented neural networks, and their analysis is also applicable to deep nets. I think the authors should provide a careful discussion of the similarity and differences.

2. In Figure 4, what is the meaning of $\min. \lambda_{\max}$?

3. I find the paragraph above Lemma 2 (the one used to motivate Lemma 2) quite confusing and the writing there can be improved.

4. In Figure 5, it is shown that the accuracy of MNIST has only around 55% for small LR but nearly 100% for large LR. This dramatical difference between small LR and large LR is very unexpected since this task is extremely simple, for which a simple logistic regression might have near 90% accuracy.  I do not understand why neural networks have only 55% accuracy even if it is trained with small LR. Is it because a large initialization is used?

**Summary Of The Paper:**

This paper studies the properties of solutions that are stable for SGD, i.e., the ones that satisfy $\lambda_1(H)\leq 2$/step size. Specifically, the authors define a stability norm and prove that the stability norm of a stable solution must be no greater than $1/\eta-1/2$. Then, the authors study the property of stability norm in both the primal space and dual space. In primal space, the stability norm can be explained as a certain weighted Laplacian norm, suggesting that training with a larger step size tends to find smoother models. The explanation in dual space is also discussed but no clear implication is made. Lastly, it is shown that this stability class can approximate functions with certain high-order smoothness.

The authors also come up with a target function, showing that this function cannot be approximated by stable two-layer neural networks but can be exactly represented by stable three-layer neural networks.


**Summary Of The Review:**

This paper provides a new analysis of how the dynamical stability of SGD can implicitly regularize the "complexity" of the learned model. The result is well supported both theoretically and empirically. I generally feel this is a good work.

---

> ### Author Response · Authors · 2022-11-14
> **Author response**
>
> We thank the reviewer for the constructive feedback.
>
> &nbsp;
>
> **Comparison to (Ma et al., 2021)** \
> As we discussed in the related work section (second paragraph), Ma & Ying (2021) studied the linear stability of SGD in expectation while considering high moments. Specifically, they introduced a necessary and sufficient condition for linear stability. Then, they combined this condition with the multiplicative structure of neural networks to prove an upper bound on the Sobolev seminorm of the model’s input-output function at stable interpolating solutions. Indeed, the Sobolev seminorm is a measure of smoothness similar to the stability norm in our paper. Yet, their upper bound depends on the norm of the first layer of the network. Namely, their result gives a bound that depends on the weights, batch size, and learning rate, whereas our bound depends only on the learning rate. Note that, in general, the norm of the first layer can be arbitrarily large, so their result might become vacuous. Following the reviewer's comment, we will extend the current discussion about this work to include more details on the similarities and differences between their work and ours.
>
> &nbsp;
>
> **The meaning of $ \\min \\lambda\_{\\max} $ in figures 4 and 5** \
> The curve $ \\min \\lambda\_{\\max} $ shows the sharpness of the flattest implementation of each solution $ f $. In more detail, Figures 4(a) and 5(a) demonstrate Theorem 1 in practice. As discussed in Sec. 3.3, Theorem 1 is a result of two properties of twice-differentiable minima. First, we know from Lemma 1 that stable minima satisfy $\\lambda\_{\\max} (\\nabla^2_\\boldsymbol{\\theta} \\mathcal{L} ) \\leq 2 / \\eta$. Second, we show in Lemma 3 in App. E that these minima satisfy $\\lambda\_{\\max} (\\nabla^2_\\boldsymbol{\\theta} \\mathcal{L} ) \\geq 1+2\\Vert f \\Vert\_{ \\mathcal{R}, g}$. Note that each function $ f \\in \\mathcal{F}\_k $ has multiple implementations, *i.e.*, different minima in parameter space which all correspond to $ f $. These minima can have different sharpness. Here, we can look at the best implementation, *i.e.* a solution to  $ \\min \\lambda\_{\\max} $, where the minimum is taken over all loss' minima $ \\{ \\boldsymbol{\\theta}  \\}  $ that implement $ f $. Overall, given a minimum $ \\boldsymbol{\\theta}  $ with a corresponding function $ f $, we have
>
> $ 1+2\\Vert f \\Vert\_{ \\mathcal{R}, g}
> \\leq \\min \\lambda\_{\\max}(\\nabla^2_\\boldsymbol{\\theta} \\mathcal{L} )
> \\leq \\lambda\_{\\max}(\\nabla^2_\\boldsymbol{\\theta} \\mathcal{L} )
> \\leq 2/\\eta  $
>
> These inequalities give us the result of Theorem 1, $1+2\\Vert f \\Vert\_{ \\mathcal{R}, g} \\leq 1/\\eta-1/2$. To understand the tightness of each part of our analysis, we added to the plots $ \\lambda\_{\\max} $ and $ \\min \\lambda\_{\\max} $. In both figures, $ \\lambda\_{\\max} (\\nabla^2_\\boldsymbol{\\theta} \\mathcal{L} ) $ equals or just below $ 2 / \\eta $, a phenomenon known as edge of stability (Cohen *et al.*, 2021). Additionally,  $ \\min \\lambda\_{\\max}(\\nabla^2_\\boldsymbol{\\theta} \\mathcal{L} ) $ is close to $ 1+2\\Vert f \\Vert\_{ \\mathcal{R}, g}  $ in these experiments. Yet, Figure 4 shows that $ \\lambda\_{\\max} (\\nabla^2_\\boldsymbol{\\theta} \\mathcal{L} ) $ can be quite larger than $ \\min \\lambda\_{\\max}(\\nabla^2_\\boldsymbol{\\theta} \\mathcal{L} ) $, meaning that there exists a far flatter minimum that implements the same function. This fact was used in (Dinh *et al.*, 2017) to show that sharp minima can generalize. We will clarify this in the final revision, thanks.
>
> &nbsp;
>
> **Clarification of the paragraph above Lemma 2** \
> Thanks for pointing this out. We clarified this in the last revision.
>
> &nbsp;
>
> **Generalization in Figure 5** \
> In Figure 5 we indeed used a large initialization ($ \\sim 10$ times larger than the default) in order to demonstrate Theorem 1 over a wide range of step sizes: from $ 10^{-4} $ to $ 1/2 $. In this setting, training with learning rates from the lower end of this spectrum leads to convergence to sharp minima, which do not generalize well. Here SGD converges at the edge of stability, and increasing the step size pushes SGD to flatter minima. We note that for deep networks in practical settings, this is the typical scenario (Cohen *et al.*, 2021). The same effect also appears when using the standard initialization, but over a more restricted range of step sizes. We will scan over different initialization scales to show this point for the final version (as these scans take time).

---

### Official Review · Reviewer_7NLR · 2022-10-22

**Confidence:** 4
**Correctness:** 4
**Technical Novelty And Significance:** 3
**Empirical Novelty And Significance:** Not applicable
**Recommendation:** 6

**Clarity, Quality, Novelty And Reproducibility:**

Clarity and quality: The paper is clear and easy to read.

Novelty: The implicit bias of minima stability for multi-dimensional two-layer neural networks is new, but built on a similar work on one-dimensional two-layer NN. The approximation of neural networks at stable minima is novel.

Reproducibility: All the experiments in the paper are on simple settings and used to justify the theory. I do not have concern on the reproducibility of the experimental results.

**Strength And Weaknesses:**

Strength:

1. The paper extends previous results on the implicit bias of minima stability for single-input two-layer neural networks to multi-input networks, making the results slightly more general.
2. The approximation capability of neural networks is studied under stability requirement. This is a more realistic setting to study the approximation properties of neural networks.

Weaknesses:

1. In the analysis of stability, only the learning rate is considered, while the batch size is ignored. Hence the stability criterion is tight only for GD. For SGD, it is just a necessary condition. The difference between GD and SGD cannot be seen in the analysis.
2. The approximation results in Section 4 and 5 are training data dependent, while in practice the training data are sampled randomly. Is it possible to develop similar results that hold with high probability over the sampling of training data?

**Summary Of The Paper:**

This paper studies the implicit bias of minima stability for two-layer neural network with multiple inputs. Based on previous works on the same topic for two-layer neural networks with single input, the authors extended the analysis to include multi-dimensional input, and showed that stability controls a weighted norm of the Laplacian of the function represented by the neural network. Discussions are made on this regularization effect in both the primal and Radon spaces. Then, the approximation capability of neural networks under stability is studied. For two-layer neural networks, it is shown that functions in a weighted Sobolev space and be approximated by networks with increasing width at stable minima. A depth separation result is proven showing that three-layer neural networks (two hidden-layer NNs) can represent a function that cannot be approximated by two-layer neural networks under any stability requirement.

**Summary Of The Review:**

This paper studies the implicit bias of minima stability and the approximation of neural networks at stable minima. The theoretical results are limited (as detailed in the "Strength and Weaknesses" part), but still interesting because (1) it accurately describes an implicit regularization effect of GD, and (2) it discusses the approximation of neural network under a more realistic setting.

---

> ### Author Response · Authors · 2022-11-14
> **Author response**
>
> We thank the reviewer for the constructive feedback.
>
> &nbsp;
>
> **The results are independent of the batch size** \
> We agree that the results are independent of the batch size and that the precise stability threshold of SGD might depend on $B$. However, empirical evidence suggests that there is not much room for improvement upon this batch-size-independent bound in common real-world settings. Specifically, for practical batch sizes, the gap between ${2}/{\\eta}$ and the stability threshold of SGD is often very small (see Fig. 5 in our paper and Figures 2-3 in (Gilmer *et al*., 2022)). Please note that the independence of Theorem 1 and Lemma 1 on the batch size is discussed in more detail in Appendix A.
>
> &nbsp;
>
> **Derivation of probabilistic versions of sections 4 and 5** \
> Interesting question. Proposition 2 (Sec. 4) states that if the support of $ p $ is contained in the interior of the convex hull of the training points, then it has an infinite stability norm. In fact, if any of the vertices of the support of $ p $ (i.e., the extreme points of the unit $\\ell^1$ ball) are contained in the interior of the convex hull of the training points this will result in an infinite stability norm. To extend this result to a probabilistic setting, it is possible to assume that the training data is sampled from some probability distribution $\\boldsymbol{x} \\sim \\mathcal{P} $ and calculate the probability that at least one of the support vertices is contained in the convex hull of the data. Exact results, however, may become highly cumbersome, even for dimension $d=2$; see for example Jewell and Romano (J. Appl. Prob 19 (1982) pp. 546-561). It is certainly possible that simple bounds on the probability can be obtained. But we leave this for future research.
>
> Regarding Section 5, it is simple to extend Theorem 2 to the probabilistic setting. In particular, this result holds with probability 1 when the training data are drawn from a distribution with compact support. More generally, extending Theorem 2 to the probabilistic setting requires extending both Proposition 3 and Lemma 2 to the probabilistic setting. For Proposition 3, this requires showing that with high probability $ c\_{d,\\hat{g}} $ is uniformly bounded, where $ \\hat{g} $ defined in Eq. 18 is now a random variable. This holds for any training set for which $ \\hat{g}(\\boldsymbol{v},b) = O(|b|) $ (as can be seen in the Proposition’s proof), *e.g.*, any probabilistic data set with compact support. Regarding Lemma 2, for any dataset with compact support, Lemma 2 will imply in high probability that $ \\lambda\_{\\max} $ is finite. Combining these two results will result in a probabilistic version of Theorem 2.

---

### Official Review · Reviewer_KQ4p · 2022-11-06

**Confidence:** 4
**Correctness:** 4
**Technical Novelty And Significance:** 4
**Empirical Novelty And Significance:** 4
**Recommendation:** 8

**Clarity, Quality, Novelty And Reproducibility:**

The quality of the work is high:
* the writing is clear
* the results are novel (although arguably they can be viewed as a natural extension of Mulayoff et al. 21, combined with Radon transform analysis from papers on representation cost: Ongie et al'20, Parhi & Nowak '21, Jin & Montufar '20)
* experiments are shown that support the relevance of the results in practice

**Strength And Weaknesses:**

### Strengths
* The proofs are clearly written and easy to follow. (I appreciated the Section C providing background on the Radon transform.)
* The analysis is novel, and significantly extends the previously known case of functions f : R \to R to functions f : R^d \to R
* The results are of interest to the neural networks community, since they give new guidance on how the step size affects the implicit bias of training.

### Weaknesses
* In Proposition 2 statement, a distribution on x should be specified?

Typos: "globaly", "can be quiet high", "this method yield", "in the space even"

**Summary Of The Paper:**

This paper generalizes the results of Mulayoff et al. 21 on the implicit bias of minima stability to functions that have multivariate output dimension. The paper considers training ReLU 2-layer networks with squared loss. The main contributions are:

* [minima stability implies smoothness] Show that linearly-stable minimizers of the loss correspond to smooth functions (where a certain weighted integral of the laplacian is not too large)

* [2-layer vs. 3-layer separation] Show that there are functions which 2-layer networks cannot represent at stable minima; but on the other hand, 3-layer networks can.

* [universal representation of smooth functions] Show that sufficiently smooth functions (in a Sobolev sense) can be approximated by stable minima.


**Summary Of The Review:**

Understanding implicit bias of neural networks is an important topic, and this paper presents significant original results. I recommend acceptance.

---

> ### Author Response · Authors · 2022-11-14
> **Author response**
>
> We thank the reviewer for the positive feedback.
>
> &nbsp;
>
> **Proposition 2 statement** \
> Proposition 2 states that the function $p(\\boldsymbol{x}) = \\text{ReLU} (1-||\\boldsymbol{x}||\_1)$ has an infinite stability norm under the assumption that the convex hull of the training points contains $ p $’s support. Here, $ x $ is just the argument of the function $p$ and is not a random variable, so no distribution should be specified. What depends on the training data in this statement, is the stability norm $\\Vert \\cdot\\Vert\_{\mathcal{R},g}$ (see its definition in Theorem 1). In our setting, we regard the training points as a given non-random dataset (note that the expectation in (12) is over the empirical distribution of this dataset, not over a theoretical population density). Therefore, the requirement in Proposition 2 that the convex-hull of the training data contains $ p $’s support, is not a random event and cannot be assigned a probability.
>
>
> If desired, it is possible to translate the statement to the setting in which the training data is drawn from some given distribution $\\boldsymbol{x} \\sim \\mathcal{P} $. In this case, the proposition would hold true under the random event that the convex-hull of the training data contains $p$’s support, whose probability depends on $\\mathcal{P} $. See our response to Reviewer 7NLR for further discussion.

---

### Decision · Program_Chairs · 2023-01-20

**Decision:**

Accept: poster

**Justification For Why Not Higher Score:**

All reviewers are favorable about the submission, but there were no advocates for a higher score.

**Justification For Why Not Lower Score:**

All reviewers are favorable about the submission. I find the article has sufficient merits to be accepted.

**Metareview: Summary, Strengths And Weaknesses:**

This work studies the stable solutions of single hidden-layer ReLU networks with multidimensional inputs trained using SGD and quadratic loss.

* Strengths are the relevance of the topic, clarity of presentation, and interesting extension of previous works, including approximation under stability requirements.
* Weaknesses are limitations of the analysis (to explain SGD and some aspects of randomness in the data), as well as restrictions to necessary conditions for stability.

The referees are all of the positive side. I agree with the indicated merits and hence recommend it to be accepted. I note there were some reservations about missing discussion of related previous works. The authors have offered to extend the current discussion and should keep this in mind when preparing the final manuscript. Following the discussion with one of the reviewers the authors also identified possibilities to construct sufficient conditions and these would further strengthen the contribution if adequately added to the final version.



**Note From Pc:**

if the above contains the word "oral" or "spotlight" please see: "oral" presentation means -> notable-top-5% and "spotlight" means -> notable-top-25%. As stated in our emails, we are disassociating presentation type from AC recommendations

**Summary Of Ac-Reviewer Meeting:**

NA